# FLATNESS-AWARE ADVERSARIAL ATTACK

## ABSTRACT

The transferability of adversarial examples can be exploited to launch black-box attacks. However, adversarial examples often present poor transferability. To alleviate this issue, by observing that the diversity of inputs can boost transferability, input regularization based methods are proposed, which craft adversarial examples by combining several transformed inputs. We reveal that input regularization based methods make resultant adversarial examples biased towards flat extreme regions. Inspired by this, we propose an attack called flatness-aware adversarial attack (*FAA*) which explicitly adds a flatness-aware regularization term in the optimization target to promote the resultant adversarial examples towards flat extreme regions. The flatness-aware regularization term involves gradients of samples around the resultant adversarial examples but optimizing gradients requires the evaluation of Hessian matrix in high-dimension spaces which generally is intractable. To address the problem, we derive an approximate solution to circumvent the construction of Hessian matrix, thereby making *FAA* practical and cheap. Extensive experiments show the transferability of adversarial examples crafted by *FAA* can be considerably boosted compared with state-of-the-art baselines.

## 1 INTRODUCTION

The transferability of adversarial examples is crucial in security-oriented applications (Liang & Xiao, 2023; Wei et al., 2023) and we explore the property on the inspiration of the following observation. Input regularization methods (Dong et al., 2019; Lin et al., 2020; Long et al., 2022) improve the transferability by attacking multiple transformed versions of inputs. We discover that such methods encourage adversarial examples to converge towards flat regions, where losses of different samples barely differ from one another. Specifically, due to the highly overlapping semantic information of transformed inputs to their original ones, transformed inputs lie in the neighborhood of the original ones. As shown in Figure 1 (left image), attacking transformed inputs can be regarded as making nearly all inputs around the original ones threatening, *i.e.*, comparable loss. In contrast, vanilla attacks (Xu et al., 2020) do not concern the neighbors of the original ones, which may cause sharp regions.

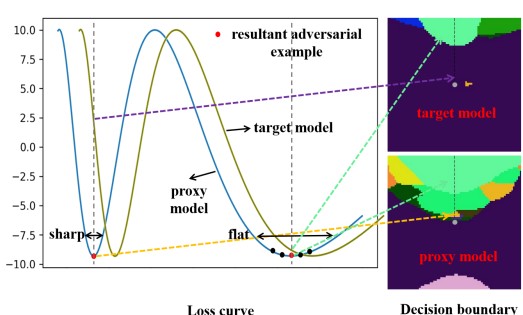

Figure 1: Different colors in the decision boundary indicate different predicted classes. A sharp region is considered a small region covered by many colors like the yellow region of the target model. Adversarial examples in flat regions (pale green regions) can be effective against both the proxy model and the target model.

In fact, the cross-model transferability is probably induced by the similar shape of the loss landscape of models associated with the input spaces; otherwise, adversarial examples produced via the proxy model should not be effective against the target model, since their loss landscapes are not correlated

and then adversarial examples should present distinct loss on different models. Furthermore, despite the overall comparability of loss landscapes among models, there still exists some discrepancy due to the non-negligible intrinsic divergence between models. To confirm the insight, we visualize the loss decision boundaries of ResNet50 (bottom) and DenseNet121 (top) and demonstrate their high similarity in loss landscapes, as shown in the right images of Figure 1. Wherein, the green regions are flat and shared by proxy and target models and the yellow, while brown regions are sharp and model-specified, which is probably caused by the unique nature of the proxy model. Hence, it is easily grasped why the adversarial examples within flat regions usually exhibit better transferability. Moreover, if attacks do not encourage adversarial examples to leap over the sharp regions, the resulting examples become trapped within sharp regions, rendering them non-transferable; vice versa.

Input regularization methods implicitly push adversarial examples towards flat regions, *i.e.*, promoting the effectiveness of surrounding ones instead of explicitly penalizing sharpness. However, implicit approaches tend to underperform explicit ones (Zhang et al., 2021; Cortes & Vapnik, 2004) due to the need for sufficiently intricate transformation techniques. The opacity of DNNs makes it intractable to identify the most effective transformation technique. To address this, we propose *FAA*, which explicitly adds a flatness-aware regularization term to encourage adversarial examples towards flat regions, together with optimizing one item for high loss. The high loss item maintains the effectiveness of the resultant adversarial examples to fool the proxy model, while the flat region item promotes them to incline towards flat regions. By jointly optimizing the two items, the produced adversarial examples are more likely to arrive at flat extreme regions, empowering better transferability.

Two challenges arise when putting our idea into practice: defining the flatness-aware regularization term and solving the optimization task. In response to the first challenge, we observe that samples located in flat regions exhibit smaller loss differences than those in sharp regions, *i.e.*, the gradients of these samples are small. Motivated by this, the flatness-aware item is defined as the norm of gradients of samples around adversarial examples. The second challenge is the prohibitive expense of constructing Hessian matrix (Fletcher, 1988) in high-dimension spaces, which is necessary to optimize the flatness-aware item. Although some approximation algorithms (Fletcher, 1988; Qian et al., 2022) have been explored to decrease overheads, they still remain costly. To tackle this, we analytically derive an approximate solution using only first-order gradients, so as to circumvent the need to directly evaluate the Hessian matrix and enable cheaper and more practical attacks. Moreover, we formally analyze the impact of the flatness-aware item on the transferability (see Appendix C), establishing a theoretical connection between transferability and flat regions. To the best of our knowledge, we are the first to demonstrate this in theory. Finally, extensive experiments show the superior performance of *FAA* over various evaluation settings. Our contribution are four-folded:

- We propose *FAA* which solves an optimization task involving two items for high loss and flat regions. Furthermore, we design an effective flatness-aware regularization item that consists of the gradient norm of samples around crafted adversarial examples.

- We derive an analytical approximation method that circumvents the need for direct computation of the Hessian matrix, thereby solving the formulated task effectively.

- We provide an theoretical analysis that illuminates the intricate relationship between flatness regions and the transferability.

- We conduct extensive experiments in both benchmark dataset ImageNet and multiple real-world applications, *e.g.*, Google Vision Systems and advanced search engines, showing the impressive performance of *FAA*. To our best knowledge, *FAA* is the first transfer-based attack to achieve an average 90% attack success rate against transformer architectures.

## 1.1 RELATED WORKS

In each iteration, input regularization methods (Dong et al., 2019; Xie et al., 2019; Wang & He, 2021; Wang et al., 2021) ensemble multiple transformed inputs to craft adversarial examples, and the distinction between these methods are reflected in the transformation techniques used. DI (Xie et al., 2019) observes the diversity of inputs can nourish the transferability and then suggests to resizes and pads the inputs to generate diverse inputs. Similarly, to obtain diverse versions of inputs, TI (Dong et al., 2019) translates the inputs and SI (Lin et al., 2020) scales the inputs. By observing that existing transformations are all applied on a single image, Admix (Wang et al., 2021) attempts to admix inputs

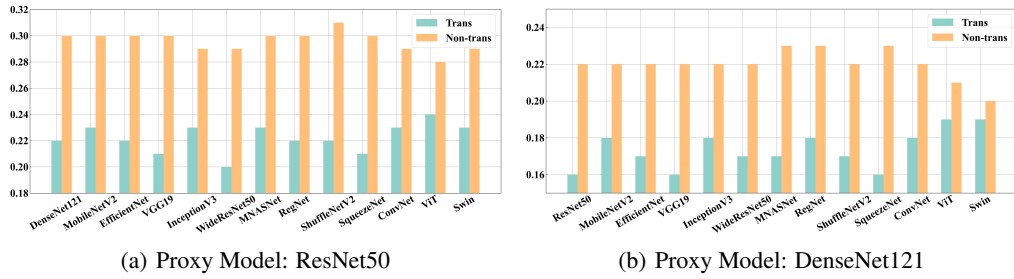

(a) Proxy Model: ResNet50          (b) Proxy Model: DenseNet121

Figure 2: The variance of model prediction for transferable and non-transferable adversarial examples. See Appendix A for more results and detailed settings.

with an image from other categories. Recently, SSA (Long et al., 2022) perturbs inputs in frequency domains to produce more diverse transformed inputs, that significantly improves the transferability.

Some methods enhance the transferability via the lens of model per se (Li et al., 2020; Zhu et al., 2022) and optimization methods (Lin et al., 2020; Dong et al., 2018). For instance, MI (Dong et al., 2018) adopts Momentum optimizer. Interestingly, it is well-shared that Momentum optimizer is capable of evading sharp extreme regions, and our insight also, to some extent, clarifies why Momentum optimizer works. Besides, SGM (Wu et al., 2020) refines the back-propagation procedure to amplify the gradients of early layers, due to that the features learned by early layers are more shared over different models. StyLess (Liang & Xiao, 2023) employs stylized networks to prevent adversarial examples from using non-robust style features.

We find that the recently proposed RAP (Qin et al., 2022) shares a similar idea that pushes adversarial examples towards flat regions. However, we distinguish this paper from Qin et al. (2022) in the following ways: 1) The motivation difference. This paper observes the connection between flat regions and input regularization methods, and RAP analogizes transferability to generalization ability of models. 2) The technical paths taken are distinct. *FAA* adds a flatness-aware item to optimization target and derive an approximation solution to address optimization. RAP formulates a SAM-like bi-level task[1]. 3) In terms of effectiveness, *FAA* enjoys time complexity of $O(n)$ while RAP has time complexity of $O(n^2)$. Experiments show higher attack success rates of *FAA* over SAM. 4) We provide a formal demonstration of the relationship between flat regions and the transferability.

## 2 LAUNCH EXPERIMENT

Before developing *FAA*, we conduct a launch experiment to validate that the adversarial examples in flat regions are more transferable. Intuitively, the predictions of the model in the presence of adversarial examples in sharp regions fluctuate more drastically than in flat regions. The fluctuation degree can be quantified by the variance of model prediction with respect to the samples in the regions. Based on this, we compute the variance of the regions of transferable and non-transferable adversarial examples respectively, thereby validating our point.

We craft adversarial examples for 10000 natural samples using BIM (Kurakin et al., 2016) on two proxy models, *i.e.*, ResNet50 and DenseNet121, with the iteration of 10 and the perturbation budget of $16/255$. We mark the adversarial examples that mislead target models (corresponding to x-axis) as Trans and the remaining ones as Non-Trans. To evaluate the flatness of regions around crafted adversarial samples, we compute variances of loss of proxy model's loss using 100 samples extracted from the neighborhood of the crafted adversarial examples. Lower variances indicate flatter regions. As shown in Figure 2, the prediction of the proxy model is less sensitive to small changes in transferable adversarial examples than non-transferable ones, *i.e.*, verifying that transferable adversarial examples fall in flat regions.

---

[1]the bi-level task also is known to be difficult to solve by common optimization methods (see Appendix B)

## 3 APPROACH

### 3.1 OPTIMIZATION FORMULATION

Given samples $x$ with ground-truth labels $y$, vanilla transfer-based adversarial attacks (Xu et al., 2020) solve the following optimization task to craft adversarial examples:

$$\delta^* = \arg\min_{\delta} -\mathcal{L}(F(x+\delta), y), \quad ||\delta||_{\infty} \leq \epsilon. \tag{1}$$

where $\mathcal{L}(\cdot, \cdot)$ is loss function and $F(\cdot)$ is the proxy model. Generally, $\mathcal{L}(\cdot, \cdot)$ is cross-entropy loss function and high loss makes $x$ to be misidentified by $F(\cdot)$. $||\delta||_{\infty} \leq \epsilon$ is the optimization constraint to make $\delta$ human-imperceptible and $\epsilon$ is a given perturbation budget. Notice that the above optimization target is just to increase the loss of the proxy model with respect to $x + \delta$. However, the resulting $x + \delta$ is easily trapped into sharp extreme points since that many sharp extreme points are around $x$ (Figure 1). To address the issue, *FAA* instead adds a flatness-aware regularization item in Equation 1 to encourage the adversarial examples to fall in flat regions. As validated in Section 2, the adversarial examples around flat regions are more transferable and, hence, the produced adversarial examples are more shared across different models.

We now turn to define our flatness-aware regularization item. In flat regions, the loss difference between points is small. Hence, the flatness of a region $U$ around $x + \delta$ can be evaluated as follows:

$$(\iint_{x_1, x_2 \in U} dx_1 dx_2)^{-1} \iint_{x_1, x_2 \in U} \frac{||\mathcal{L}(F(x_1), y) - \mathcal{L}(F(x_2), y)||}{||x_1 - x_2||} dx_1 dx_2. \tag{2}$$

Equation 2 measures the average slope of any two points in the region $U$. However, directly optimizing Equation 2 is expensive since it requires enumerating all possible pairs of $x_1$ and $x_2$. Fortunately, based on Mean Value Theorem (Stein & Shakarchi, 2005), there is $\frac{||\mathcal{L}(F(x_1), y) - \mathcal{L}(F(x_2), y)||}{||x_1 - x_2||} = ||\nabla\mathcal{L}(F(\xi), y)||$, where $\xi$ is between $x_1$ and $x_2$. If $||\nabla\mathcal{L}(F(x), y)|| \leq k$ for $\forall x \in U$, there is an upper bound of Equation 2:

$$(\iint_{x_1, x_2 \in U} dx_1 dx_2)^{-1} \iint_{x_1, x_2 \in U} \frac{||\mathcal{L}(F(x_1), y) - \mathcal{L}(F(x_2), y)||}{||x_1 - x_2||} dx_1 dx_2$$
$$\leq (\iint_{x_1, x_2 \in U} dx_1 dx_2)^{-1} \iint_{x_1, x_2 \in U} k \ dx_1 dx_2 = k. \tag{3}$$

Therefore, penalizing the gradients of samples around $x + \delta$ can encourage the resultant $x + \delta$ to fall in flat regions. Compared to directly penalizing Equation 2, penalizing the gradients only involves a single variable, significantly reducing the optimization costs required. In practice, *FAA* penalizes the gradients of $N$ samples uniformly extracted from the small region around $x + \delta$ to conduct unbiased monte carlo estimation. Specifically, the optimization target of *FAA* is formulated as follows:

$$\delta^* = \arg\min_{\delta} - \mathcal{L}(F(x+\delta), y) + \frac{\lambda}{N} \sum_{i=1}^{N} ||\nabla\mathcal{L}(F(x+\delta+\Delta_i), y)||_2,$$
$$||\delta||_{\infty} \leq \epsilon, \Delta_i \sim U(-b, b), b \geq 0, \tag{4}$$

where $U(-b, b)$ is uniform distribution between $-b$ and $b$, $\lambda$ is penalty magnitude, and $x + \delta + \Delta_i$ denotes a sample extracted uniformly from the region around $x + \delta$. $b$ determines how wide the region around $x + \delta$ is desired to be flat, and higher $\lambda$ suggests more attention to be paid to searching flat regions. Moreover, intuitively, by incorporating a gradient penalty term into Equation 4, the generated adversarial examples become less sensitive to changes in models. Thus, the resulting adversarial examples probably enjoy small loss change on similar models, *i.e.*, high transferability. See Appendix C for the formal discussion.

### 3.2 APPROXIMATE SOLUTION

The highly non-linear and non-convex nature of $F(\cdot)$ render the analytic solution of Equation 4 to be hardly derived (Kurakin et al., 2016). As a result, standard practice for solving Equation 4 is employing gradient-based optimization methods (Croce & Hein, 2020). However, as shown in

Equation 5, the gradients of Equation 4 involve Hessian matrix evaluated at multiple points $x+\delta+\Delta_i$ for $\Delta_i, i = 1, \cdots, N$, each of which is troublesome to evaluate in high-dimension spaces (Fletcher, 1988; Qian et al., 2022).

$$- \nabla\mathcal{L}(F(x+\delta),y) + \frac{\lambda}{N}\sum_{i=1}^{N}\nabla||\nabla\mathcal{L}(F(x+\delta+\Delta_i),y)||_2$$
$$= -\nabla\mathcal{L}(F(x+\delta),y) + \frac{\lambda}{N}\sum_{i=1}^{N}H_{x+\delta+\Delta_i}\frac{\nabla\mathcal{L}(F(x+\delta+\Delta_i),y)}{||\nabla\mathcal{L}(F(x+\delta+\Delta_i),y)||_2}. \tag{5}$$

To get rid of directly evaluating Hessian matrix in Equation 5, we propose an approximate estimation for $H_{x+\delta+\Delta_i}\frac{\nabla\mathcal{L}(F(x+\delta+\Delta_i),y)}{||\nabla\mathcal{L}(F(x+\delta+\Delta_i),y)||_2}$. To achieve this, we utilize Taylor expansion on $\mathcal{L}(F(x+\delta+\Delta_i+\phi),y)$, assuming $\phi$ is small enough for making expansion feasible:

$$\mathcal{L}(F(x+\delta+\Delta_i+\phi),y) = \mathcal{L}(F(x+\delta+\Delta_i),y) + \nabla\mathcal{L}(F(x+\delta+\Delta_i),y)\phi, \tag{6}$$

Furthermore, by differentiating both sides of Equation 6, we obtain:

$$\nabla\mathcal{L}(F(x+\delta+\Delta_i+\phi),y) = \nabla\mathcal{L}(F(x+\delta+\Delta_i),y) + H_{x+\delta+\Delta_i}\phi, \tag{7}$$

Notice that, the desired item $H_{x+\delta+\Delta_i}\frac{\nabla\mathcal{L}(F(x+\delta+\Delta_i),y)}{||\nabla\mathcal{L}(F(x+\delta+\Delta_i),y)||_2}$ arises if setting $\phi$ along the direction $\frac{\nabla\mathcal{L}(F(x+\delta+\Delta_i),y)}{||\nabla\mathcal{L}(F(x+\delta+\Delta_i),y)||_2}$. By implementing the idea, there is:

$$H_{x+\delta+\Delta_i}\frac{\nabla\mathcal{L}(F(x+\delta+\Delta_i),y)}{||\nabla\mathcal{L}(F(x+\delta+\Delta_i),y)||_2} =$$
$$\underbrace{\frac{\nabla\mathcal{L}(F(x+\delta+\Delta_i+k\frac{\nabla\mathcal{L}(F(x+\delta+\Delta_i),y)}{||\nabla\mathcal{L}(F(x+\delta+\Delta_i),y)||_2}),y)}{k}}_{Term\ 1} - \underbrace{\frac{\nabla\mathcal{L}(F(x+\delta+\Delta_i),y)}{k}}_{Term\ 2}. \tag{8}$$

Wherein, $\phi = k\frac{\nabla\mathcal{L}(F(x+\delta+\Delta_i),y)}{||\nabla\mathcal{L}(F(x+\delta+\Delta_i),y)||_2}$ and $k$ is a small constant to cater $\phi$ being small. By substituting Equation 8 into Equation 5, we obtain the approximately estimated gradients of Equation 4. Compared to directly evaluating Hessian matrix that requires quadratic storage and cubic computation time (Fletcher, 1988; Qian et al., 2022), our solution only involves first-order gradients (linear computation time) so as to make *FAA* more efficient computationally. Moreover, linear expansion used in Equation 7 results in an approximation error of $O(k)$ (Stein & Shakarchi, 2005). If $L(F(\cdot),\cdot)$ is $\psi$-Lipschitz continuous Hessian, the approximation error is lower than $\frac{\psi k}{2}$. See Appendix D for empirical error evaluation.

### 3.3 A TOY EXAMPLE

We construct a objective function $\frac{sin(\frac{1}{x})}{(x-0.15)^2+0.1} + 100(x - 0.15)^2$ involving sharp and flat extreme regions to exemplify that *FAA* can find flat regions. We employ SGD optimizer and *FAA* with learning rate of 0.01 and random initialization strategy ($U(0.075, 0.250)$) to search for the best solutions that minimize the objective function. Both SGD and *FAA* are run 1000 times each. Out of these runs, SGD converge to a flat region (around 0.2) only 649 times, while *FAA* find such regions 948 times, showing the superiority of *FAA* to locate flat areas.

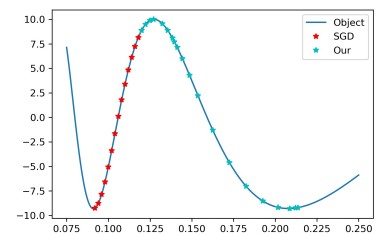

Figure 3: The search trajectories of SGD and our method *FAA*.

Figure 3 shows an example, of the search trajectories using different optimizers with initialization point of 0.01. Wherein, SGD optimizer obtains the best solutions of about 0.1. However, the right extreme point, situated in a flatter region, is more favorable compared to the left extreme point. We employ *FAA* to search for the best solution and the obtained solution is convergent at the right extreme point, indicating that *FAA* is indeed aware of the flatness of surrounding regions and favors a solution centered around flat regions.

Table 1: The attack success rates (%) of different attacks on normal models. Best results are in bold.

| Proxy Model | Method | ResNet50 | DenseNet121 | EfficientNet | InceptionV3 | MobileNetV2 | SqueezeNet | ShuffleNetV2 | ConvNet | RegNet | MNASNet | WideResNet50 | VGG19 | ViT | Swin |
|---|---|---|---|---|---|---|---|---|---|---|---|---|---|---|---|
| ResNet50 | BIM | **100.00** | 40.48 | 34.31 | 38.62 | 41.30 | 53.62 | 44.08 | 26.35 | 34.03 | 42.63 | 41.51 | 42.98 | 24.59 | 27.14 |
| | MI | **100.00** | 87.91 | 76.43 | 68.57 | 66.51 | 86.66 | 78.11 | 60.84 | 72.50 | 81.33 | 90.77 | 81.99 | 43.06 | 55.89 |
| | NI | **100.00** | 88.14 | 76.79 | 68.57 | 66.10 | 86.43 | 78.06 | 60.64 | 72.55 | 82.35 | 91.17 | 82.98 | 41.68 | 54.87 |
| | DI | 99.97 | 80.77 | 68.75 | 69.74 | 66.05 | 77.58 | 72.83 | 49.39 | 65.82 | 79.54 | 82.91 | 76.51 | 40.33 | 46.71 |
| | TI | **100.00** | 73.47 | 56.28 | 55.99 | 55.79 | 68.85 | 62.04 | 42.83 | 59.08 | 66.33 | 78.60 | 68.78 | 34.85 | 40.51 |
| | VT | **100.00** | 89.62 | 81.63 | 73.47 | 76.86 | 90.21 | 83.83 | 76.28 | 82.83 | 92.86 | 95.51 | 86.12 | 54.77 | 57.87 |
| | SSA | **100.00** | 95.51 | 91.56 | 79.62 | 77.40 | 93.60 | 87.60 | 79.90 | 85.99 | 93.60 | 96.53 | 95.13 | 54.21 | 59.64 |
| | RAP | **100.00** | 95.01 | 94.78 | 93.81 | 93.88 | 93.87 | 94.64 | 90.80 | 94.54 | 95.25 | 93.97 | 94.92 | 62.62 | 60.77 |
| | StyLess | **100.00** | 98.16 | 95.75 | 94.21 | 94.98 | 95.12 | 95.63 | 89.19 | 95.77 | 97.20 | 94.10 | 94.66 | 83.27 | 79.43 |
| | Ours | 99.85 | **99.69** | **99.52** | **98.70** | **99.52** | **99.72** | **99.82** | **94.34** | **97.37** | **99.82** | **99.44** | **98.83** | **93.52** | **90.98** |
| ResNet152 | BIM | 45.77 | 40.08 | 34.57 | 38.83 | 40.64 | 53.37 | 43.78 | 26.48 | 34.85 | 41.99 | 41.56 | 41.96 | 24.72 | 27.14 |
| | MI | 83.44 | 85.92 | 77.70 | 70.97 | 66.33 | 85.54 | 77.65 | 62.17 | 74.59 | 81.30 | 90.36 | 79.52 | 45.43 | 56.22 |
| | NI | 84.26 | 87.32 | 77.70 | 70.38 | 66.38 | 86.10 | 78.06 | 62.88 | 74.67 | 81.48 | 90.89 | 78.70 | 43.93 | 54.31 |
| | DI | 86.76 | 79.06 | 70.56 | 71.38 | 66.89 | 75.87 | 73.06 | 52.70 | 68.75 | 78.67 | 83.01 | 73.80 | 43.62 | 47.45 |
| | TI | 84.21 | 73.42 | 58.57 | 57.60 | 56.51 | 68.14 | 62.42 | 45.08 | 61.35 | 66.48 | 78.57 | 66.48 | 37.91 | 41.30 |
| | VT | 89.54 | 89.57 | 84.67 | 83.62 | 77.98 | 92.07 | 87.76 | 72.42 | 89.29 | 92.5 | 91.38 | 90.87 | 56.38 | 55.87 |
| | SSA | 94.62 | 95.23 | 93.06 | 88.55 | 78.55 | 96.05 | 93.37 | 76.79 | 93.11 | 93.24 | 94.74 | 93.11 | 57.63 | 59.46 |
| | RAP | 98.86 | 94.44 | 93.74 | 93.48 | 93.61 | 92.37 | 93.07 | 90.01 | 92.89 | 94.82 | 93.22 | 94.22 | 61.48 | 58.98 |
| | StyLess | 99.10 | 95.54 | 94.78 | 95.11 | 94.48 | 93.57 | 94.88 | 91.22 | 93.25 | 95.10 | 95.10 | 95.46 | 83.48 | 78.54 |
| | Ours | **99.85** | **99.67** | **99.39** | **98.95** | **99.18** | **99.01** | **99.64** | **94.11** | **97.96** | **99.77** | **99.57** | **98.57** | **90.19** | **86.27** |
| DenseNet121 | BIM | 42.07 | **100.00** | 33.67 | 38.75 | 41.76 | 54.08 | 44.26 | 25.99 | 32.50 | 41.96 | 37.96 | 42.91 | 24.39 | 27.27 |
| | MI | 89.01 | 99.97 | 81.35 | 73.95 | 70.33 | 87.88 | 80.18 | 65.00 | 73.65 | 84.41 | 86.63 | 85.64 | 47.17 | 61.10 |
| | NI | 90.28 | 99.98 | 82.55 | 74.13 | 70.10 | 87.65 | 81.15 | 65.51 | 74.26 | 86.10 | 87.98 | 86.28 | 47.60 | 61.45 |
| | DI | 82.45 | 99.90 | 73.24 | 72.88 | 70.43 | 78.90 | 76.58 | 51.33 | 66.22 | 81.22 | 78.47 | 79.52 | 43.52 | 49.92 |
| | TI | 76.33 | 99.95 | 61.38 | 60.15 | 59.01 | 70.41 | 66.33 | 46.15 | 58.78 | 70.84 | 72.76 | 73.60 | 38.88 | 44.21 |
| | VT | 92.12 | 99.95 | 90.33 | 86.58 | 81.48 | 92.02 | 90.18 | 75.19 | 85.78 | 78.75 | 81.20 | 81.56 | 49.18 | 52.93 |
| | SSA | 94.82 | 99.97 | 91.86 | 83.67 | 83.67 | 96.66 | 92.76 | 82.18 | 88.04 | 88.11 | 90.48 | 90.13 | 50.92 | 59.64 |
| | RAP | 97.40 | 99.96 | 95.48 | 95.06 | 94.67 | 97.91 | 94.94 | 90.77 | 92.51 | 96.10 | 95.04 | 94.43 | 63.13 | 61.65 |
| | StyLess | 99.19 | 99.98 | 96.24 | 95.49 | 95.50 | 97.85 | 96.13 | 91.22 | 94.34 | 96.58 | 90.96 | 96.82 | 83.76 | 81.42 |
| | Ours | **99.69** | 99.87 | **99.34** | **98.32** | **99.52** | **99.57** | **99.82** | **94.41** | **97.37** | **99.77** | **99.08** | **99.03** | **93.70** | **91.98** |
| DenseNet201 | BIM | 43.32 | 46.81 | 35.31 | 39.41 | 41.25 | 53.49 | 44.97 | 27.09 | 33.65 | 42.76 | 39.23 | 42.07 | 25.23 | 28.06 |
| | MI | 90.41 | 95.89 | 83.16 | 74.90 | 70.56 | 87.53 | 80.69 | 69.21 | 76.33 | 85.08 | 87.60 | 83.34 | 51.28 | 64.31 |
| | NI | 91.20 | 96.56 | 83.19 | 75.48 | 70.20 | 87.32 | 80.87 | 70.18 | 77.24 | 86.25 | 88.55 | 84.67 | 50.48 | 64.54 |
| | DI | 84.57 | 90.28 | 76.02 | 75.71 | 70.74 | 77.68 | 77.76 | 56.71 | 70.99 | 82.53 | 81.38 | 78.57 | 47.78 | 53.67 |
| | TI | 78.39 | 88.11 | 63.85 | 62.37 | 59.67 | 70.00 | 66.61 | 49.21 | 63.29 | 71.30 | 74.11 | 70.74 | 41.53 | 46.38 |
| | VT | 93.44 | 95.91 | 83.06 | 87.40 | 80.71 | 89.92 | 93.04 | 79.92 | 84.89 | 79.16 | 82.12 | 79.16 | 42.37 | 55.66 |
| | SSA | 96.20 | 97.27 | 92.42 | 84.11 | 82.88 | 94.59 | 90.31 | 77.93 | 90.33 | 87.42 | 89.82 | 87.70 | 42.50 | 62.12 |
| | RAP | 96.01 | 98.99 | 94.42 | 95.15 | 93.27 | 97.56 | 93.58 | 89.05 | 91.04 | 95.64 | 93.74 | 92.52 | 62.66 | 61.31 |
| | StyLess | 98.26 | 99.56 | 95.77 | 96.42 | 94.72 | 97.84 | 94.76 | 93.44 | 93.57 | 96.75 | 95.81 | 93.57 | 82.54 | 80.15 |
| | Ours | **99.72** | **99.90** | **99.67** | **98.83** | **99.23** | **99.31** | **99.85** | **96.96** | **98.34** | **99.77** | **99.62** | **99.16** | **93.54** | **92.85** |

Here we give insights into the effectiveness of *FAA*. Consider the situation that the optimization variable falls in flat regions. In flat regions, the gradients of nearby points show small variation, causing the flatness-aware regularization term (estimated by Equation 8) to close to 0. Thus, the high loss item dominates Equation 5 and *FAA* can be deemed to degrade into vanilla transfer-based attacks.

On the contrary, if optimization variables falls in a sharp region, $x + \delta + \Delta_i + k\frac{\nabla\mathcal{L}(F(x+\delta+\Delta_i),y)}{||\nabla\mathcal{L}(F(x+\delta+\Delta_i),y)||_2}$ probably goes beyond the sharp region due to considerable gradients of points over sharp regions. In this way, *Term 1* in Equation 8 informs gradient information about other regions. Moreover, the gradient directions of points over a small neighborhood of $x + \delta$ should share similar directions. Hence, *Term 2* in Equation 8 in fact weakens the impact of the high loss item. In summary, in this situation, *FAA* will encourage the optimization variables to explore more regions.

## 4 SIMULATION EXPERIMENT

### 4.1 SETUP

**Dataset.** We randomly select 10000 images from the validation set of ImageNet. These images span the whole label domain of ImageNet (1000 classes).

**Models.** We consider 14 models including ResNet50, DenseNet121, MobileNetV2, EfficientNet, VGG19, InceptionV3, WideResNet50, MNASNet, RegNet, ShuffleNetV2, SqueezeNet, ConvNet, ViT, and Swin. The models cover mainstream CV models: the former 12 are convolutional networks and the last 2 are transformer-like networks. Moreover, we also validate the performance of *FAA* on secured models including adversarial training with $L_2$ (Salman et al., 2020) and $L_\infty$ (Tramèr et al., 2018) constraint as well as robust training with Styled ImageNet (SIN) and the mixture of Styled and natural ImageNet (SIN-IN) (Geirhos et al., 2019).

**Baselines.** Nine state-of-the-art attacks are used as competitors for *FAA*: BIM (Kurakin et al., 2016), DI (Xie et al., 2019), MI (Dong et al., 2018), NI (Lin et al., 2020), TI (Dong et al., 2019), VT (Wang & He, 2021), SSA (Long et al., 2022), RAP (Qin et al., 2022), StyLess (Liang & Xiao, 2023), and Self-Universality (Wei et al., 2023). Self-Universality is a targeted attack and thus we only report its performance in targeted setting (Table 3).

**Evaluation metric.** Attack success rate is used as the evaluation metric that is defined as the misclassified rate of adversarial examples by target models. The higher the attack success rate, the better the attack performance.

Table 2: The attack success rates (%) of different methods on secured models. Three different robust training methods are considered: adversarial training with $L_2$ perturbation ($L2 - \{0.03 \sim 5\}$) (Salman et al., 2020) and $L_\infty$ perturbation (AdvIncV3 and EnsAdvIncResV2) (Tramèr et al., 2018), robust training with Styled ImageNet (SIN) and the mixture of Styled and natural ImageNet (SIN-IN) (Geirhos et al., 2019). The best results are in bold.

| Proxy Model | Method | AdvIncV3 | EnsAdvIncResV2 | SIN | SIN-IN | L2-0.03 | L2-0.05 | L2-0.1 | L2-0.5 | L2-1 | L2-3 | L2-5 |
|---|---|---|---|---|---|---|---|---|---|---|---|---|
| ResNet50 | BIM | 41.81 | 33.52 | 50.51 | 51.56 | 63.67 | 64.06 | 62.81 | 61.10 | 63.98 | 68.27 | 74.13 |
| | MI | 58.78 | 43.24 | 75.46 | 96.05 | 82.83 | 83.47 | 80.31 | 74.74 | 75.00 | 75.08 | 77.98 |
| | NI | 58.11 | 43.11 | 75.13 | 96.79 | 83.57 | 83.70 | 79.97 | 74.52 | 75.15 | 75.33 | 78.11 |
| | DI | 54.34 | 45.03 | 75.84 | 92.04 | 78.27 | 78.19 | 75.61 | 69.57 | 70.94 | 71.53 | 75.74 |
| | TI | 48.52 | 39.72 | 66.35 | 88.27 | 75.51 | 75.05 | 72.76 | 67.04 | 69.21 | 70.87 | 75.61 |
| | VT | 59.72 | 50.89 | 69.03 | 94.06 | 77.27 | 77.22 | 74.21 | 68.24 | 69.80 | 70.92 | 75.48 |
| | SSA | 60.13 | 50.48 | 70.31 | 97.63 | 80.61 | 80.46 | 76.84 | 69.06 | 70.28 | 71.25 | 75.69 |
| | Ours | **74.57** | **72.81** | **99.52** | **99.90** | **96.20** | **96.45** | **96.48** | **93.65** | **91.76** | **88.19** | **87.24** |
| ResNet152 | BIM | 41.84 | 33.85 | 48.95 | 42.14 | 63.11 | 63.47 | 62.76 | 60.92 | 64.11 | 68.19 | 74.06 |
| | MI | 58.78 | 45.15 | 71.40 | 82.58 | 82.55 | 82.96 | 80.13 | 74.87 | 75.23 | 75.36 | 78.21 |
| | NI | 58.98 | 44.74 | 71.79 | 83.83 | 82.70 | 83.04 | 80.08 | 74.80 | 75.13 | 75.28 | 78.14 |
| | DI | 54.97 | 47.58 | 72.50 | 80.00 | 77.27 | 76.86 | 75.38 | 69.46 | 70.26 | 71.56 | 75.59 |
| | TI | 49.34 | 41.48 | 63.11 | 72.76 | 74.21 | 74.44 | 72.22 | 66.61 | 69.06 | 71.35 | 75.48 |
| | VT | 51.22 | 43.42 | 65.10 | 78.62 | 76.58 | 76.07 | 73.44 | 68.11 | 69.72 | 71.02 | 75.41 |
| | SSA | 51.30 | 41.94 | 67.68 | 84.62 | 79.34 | 79.31 | 76.61 | 69.29 | 70.20 | 71.20 | 75.54 |
| | Ours | **71.66** | **79.46** | **99.21** | **99.80** | **95.64** | **95.43** | **95.15** | **91.76** | **90.05** | **85.36** | **84.72** |
| DenseNet121 | BIM | 41.99 | 34.08 | 48.80 | 40.26 | 63.09 | 63.11 | 62.55 | 60.92 | 64.01 | 68.27 | 74.16 |
| | MI | 59.31 | 45.03 | 71.25 | 79.44 | 81.71 | 83.27 | 80.08 | 74.64 | 76.02 | 75.77 | 78.65 |
| | NI | 59.03 | 45.08 | 72.07 | 80.31 | 81.53 | 83.14 | 79.80 | 74.87 | 75.69 | 75.79 | 78.57 |
| | DI | 55.00 | 47.19 | 73.21 | 77.19 | 76.99 | 77.07 | 75.94 | 69.31 | 70.69 | 71.68 | 75.69 |
| | TI | 49.13 | 40.71 | 62.24 | 67.58 | 73.62 | 73.57 | 71.96 | 67.60 | 68.85 | 71.10 | 75.38 |
| | VT | 60.26 | 51.73 | 64.26 | 75.28 | 76.28 | 76.38 | 74.21 | 68.55 | 69.97 | 71.05 | 75.54 |
| | SSA | 61.84 | 51.86 | 69.08 | 84.06 | 79.64 | 79.85 | 76.94 | 69.67 | 70.51 | 71.45 | 75.77 |
| | Ours | **72.19** | **79.64** | **99.16** | **99.64** | **95.31** | **95.69** | **95.56** | **92.65** | **90.82** | **86.63** | **85.71** |
| DenseNet201 | BIM | 42.07 | 34.18 | 48.93 | 40.99 | 62.88 | 63.62 | 62.78 | 60.94 | 64.11 | 68.42 | 74.13 |
| | MI | 59.92 | 45.20 | 71.33 | 81.53 | 82.98 | 83.19 | 80.03 | 75.54 | 76.20 | 75.71 | 78.60 |
| | NI | 59.90 | 44.41 | 72.27 | 82.42 | 82.55 | 83.06 | 80.18 | 75.28 | 75.79 | 75.66 | 78.21 |
| | DI | 56.53 | 48.70 | 74.69 | 80.03 | 77.12 | 77.45 | 76.73 | 70.05 | 71.43 | 71.71 | 75.82 |
| | TI | 50.08 | 41.20 | 62.68 | 69.34 | 74.21 | 74.29 | 72.60 | 67.68 | 69.67 | 71.02 | 75.64 |
| | VT | 61.76 | 52.86 | 65.23 | 76.51 | 76.66 | 76.86 | 74.59 | 68.67 | 70.26 | 71.07 | 75.64 |
| | SSA | 62.30 | 53.76 | 69.59 | 85.05 | 79.64 | 79.39 | 76.76 | 70.20 | 70.92 | 71.38 | 75.99 |
| | Ours | **70.99** | **78.21** | **98.98** | **99.67** | **95.41** | **95.94** | **94.92** | **91.56** | **89.44** | **85.51** | **84.57** |

Table 3: The targeted attack success rates of different methods. The proxy model is ResNet50.

| Method | DenseNet121 | EfficientNet | InceptionV3 | ConvNet | WideResNet50 | VGG19 | ViT | Swin |
|---|---|---|---|---|---|---|---|---|
| MI | 4.20 | 0.60 | 0.00 | 0.20 | 5.20 | 0.80 | 0.00 | 0.20 |
| DI | 1.60 | 0.20 | 0.20 | 0.00 | 2.00 | 0.70 | 0.00 | 0.10 |
| VT | 4.74 | 1.08 | 0.53 | 1.05 | 5.43 | 1.31 | 0.42 | 0.38 |
| SSA | 5.39 | 2.27 | 1.43 | 0.70 | 6.65 | 3.56 | 0.81 | 1.20 |
| RAP | 18.64 | 13.90 | 8.23 | 6.76 | 12.20 | 7.10 | 7.30 | 5.82 |
| StyLess | 19.85 | 14.27 | 9.15 | 8.50 | 14.85 | 9.67 | 7.96 | 6.75 |
| Self-Universality | 22.00 | 18.04 | 9.59 | 8.69 | 15.69 | 8.52 | 9.72 | 6.69 |
| Ours | **31.90** | **25.20** | **14.30** | **12.20** | **22.60** | **13.20** | **13.40** | **9.80** |

**Hyperparameter configurations.** For baselines, we set hyperparameters used in their original papers by default. For *FAA*, we set $\lambda = 5, b = 16, k = 0.05, N = 10$. Moreover, for all methods, we set iteration of 20, $\epsilon$ of 16, and step size of 1.6.

See supplement materials for source codes. See Appendix E.1, E.3, E.4, E.5, E.6, E.7, E.8 for attack results in other kinds of defenses like NRP (Naseer et al., 2020), the impact of hyperparameters, $L_2$-norm constraint attack, the impact of iterations, comparsion with transformer-based attacks, the generality validation of *FAA*, comparison of flatness between adversarial examples generated by *FAA* and RAP. In the following parts, our primary focus is on SSA as it stands out as state-of-the-art input regularization attack.

## 4.2 ATTACK RESULTS

**Attack results on normal models.** Here we examine the attack effectiveness of our method on both convolutional neural networks and transformer-like neural networks and Table 1 reports the attack results. The following observation can be made. Simply speaking, as shown in Table 1, our method defeats all the baselines by a significant margin and is the winning attack. For instance, when employing ResNet50 as the proxy model, our attack witnesses an average gain of roughly 9% on attack success rate, even when compared to the state-of-the-art transfer-based attack SSA.

**Attack results on secured models.** We evaluate the effectiveness of *FAA* on models defended by robust training. We stick to employing undefended models as proxy models and this is a more challenging setting, given that proxy and target models are more divergent. The attack results are reported in Table 2 and we draw the following conclusions. Similar to the attack results on normal

models, *FAA* still can craft more transferable adversarial examples against robust models. In particular, as can be seen in Table 2, *FAA* enhance the attack success rates over SSA by a large margin.

**Targeted attack results.** We also investigate the attack effectiveness of *FAA* for targeted attacks, which is a more challenging attack setting compared to untargeted attack setting. Following the evaluation setting for targeted attacks (Zhao et al., 2021), Table 3 reports the targeted attack results of different attack methods. As can be seen in Table 3, the attack effectiveness of *FAA* on the targeted setting is so striking, even when compared to Self-Universality, which is specifically designed for targeted attacks. compared with baselines.

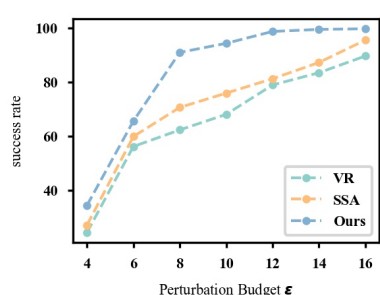

**Attack results with varying perturbation budgets.** Small perturbation budgets $\epsilon$ raise the difficulty in conducting transfer-based attacks and we further evaluate the attack effectiveness of different methods with varying perturbation budgets. Figure 4 illustrates the

Figure 4: The impact of perturbation budgets on the effectiveness of *FAA* using a Proxy Model of ResNet50.

attack success rates of VT, SSA, and *FAA*. In short, *FAA* still dominates all settings in terms of attack success rates.

### 4.3 A Closer Look at *FAA*

We are also interested in how produced adversarial noises fool target models. Figure 5 visualizes the generated adversarial noises generated by VT, SSA, and *FAA*, respectively. Overall, the adversarial noises with VT and SSA unevenly spread throughout the entire images, whereas the adversarial noises with *FAA* seem to be more semantic and focus on tampering with the part of the entity in the images, *i.e.*, the dog, car, and tree. Intuitively, tampering with the features of the entities is more likely to make the resultant adversarial examples transferable, since models indeed use the features of the entities to make predictions. Moreover, we examine whether or not the adversarial noises can lure the attention of the target model (DenseNet121) to deviate from the location of the target class (dog). The bottom images of Figure 5 verify our point that the attention of the target model focuses on the upper area rather than the location of the target class. Interestingly, we find that the noises in

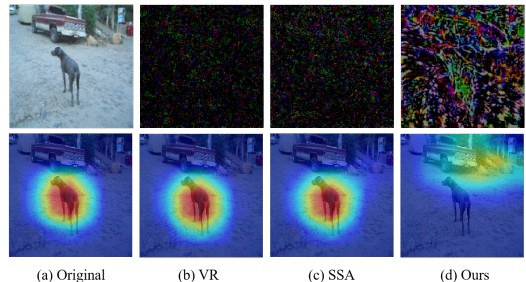

(a) Original     (b) VR     (c) SSA     (d) Ours

Figure 5: The upper images visualize the crafted noises using different methods while the below images show the attention maps of the target model to corresponding adversarial examples by employing GradCam (Selvaraju et al., 2017). We provide visualizations for targeted attacks in Appendix E.2.

the upper area of produced adversarial noises with *FAA* are intensive and the target model also paid more attention to the area than the unperturbed version. This leads to speculation that *FAA* can distinguish different entities in the images, corrupt the features of the entity matching the ground-truth label and reinforce the features of leaving entities, so as to fool the target model.

## 5 Evaluation in Real CV Applications

This section evaluates the performance of *FAA* against real deployed computer vision systems, which is more challenging but also leads to a more reliable evaluation due to the following three reasons: 1) **Complexity and architecture of the target applications.** The model used in real applications is unknown and the model can range from a simple network to an extremely sophisticated

Table 4: The scoring for the effectiveness of adversarial examples against real-world applications. We generate 100 adversarial examples using *FAA* and then enlist a volunteer to assess the consistence between the image contents with the predictions made by applications. A lower rating reflects a higher effectiveness of the attack. A rating of 1 signifies that adversarial examples completely fool the applications to a significant extent. The results of other attacks can be found in Appendix E.9.

| Score | Classification | Object Detection | Google Search Engine | Bing Search Engine | Yandex Search Engine | Baidu Search Engine |
|---|---|---|---|---|---|---|
| 5 | 1 | 3 | 0 | 0 | 0 | 0 |
| 4 | 7 | 21 | 10 | 6 | 5 | 4 |
| 3 | 13 | 7 | 18 | 11 | 13 | 4 |
| 2 | 9 | 4 | 16 | 21 | 17 | 10 |
| 1 | 70 | 65 | 56 | 62 | 65 | 82 |

combination of various networks. Besides, such applications may be equipped with some practical defenses. 2) **Training setting.** Publicly available models mostly share similar training settings (ImageNet). However, the settings of real industry environments probably are far more complicated. 3) **Structure of the output.** Real-world systems often output multiple hierarchical labels with associated confidences, instead of logits. Besides, the label domain of the target applications is extremely larger than that of proxy models, i.e., inconsistency in label space.

**Google MLaaS platforms.** We attack Google Cloud Vision Application [2] including image classification and object detection, which is believed to be one of the most advanced AI services. We craft adversarial ones for 100 images against the applications. For image classification and object detection, we collect predictions made by the Google Vision System and seek a volunteer[3] to score the consistency of the adversarial ones with the corresponding predictions. The scoring ranges from 1 (totally wrong) to 5 (precise), with higher scores indicating weaker attack performance. See the appendix F for further details about the scoring process. Table 4 shows the striking effectiveness of *FAA* against Google Service, while Appendix G gives the visualization results. The attack success rates for image classification and object detection are around 70% and 80%, respectively, when considering a score of 1 or 2 as a successful attack. Besides, we find that ineffective adversarial examples often involve entities of persons, which ImageNet does not contain as a label. As a result, *FAA* is not guided to corrupt the features of "Person," leading to the ineffectiveness of these adversarial examples.

**Reverse image search engine.** Given an image of interest, reverse image search engine enables searching for the most similar ones and creates great convenience and benefits. The task is remarkably different from image classification and object detection and here we test the effectiveness of *FAA* against the engines.

Google, Bing, Yandex, and Baidu Picture Search are Top-4 search engines suggested by the site[4]. We reuse adversarial ones created for Google Service and score their effectiveness ranging from 1 to 5. The attack effectiveness is negatively correlated with the similarity between the original image and the images retrieved by the engines. Table 4 reports attack results and Appendix G shows the images retrieved by four search engines for original and adversarial ones. Four engines present notable vulnerabilities against adversarial examples by *FAA*. Specifically, the retrieved images for adversarial ones are significantly inferior to those for original images, particularly for Baidu Picture Search, which returns completely unrelated images.

## 6 CONCLUSION

We proposed *FAA* that involves a flatness-aware regularization item to encourage crafted adversarial examples towards flat regions so as to boost the transferability of crafted adversarial examples. We designed a flat item that consists of the gradients of samples around the crafted adversarial examples and derived an approximate solution to circumvent the construction of Hessian matrix, making *FAA* cheaper and more practical. We conducted extensive experiments in ImageNet over various proxy-target model pairs and real-world CV applications and the results showed the superior effectiveness of *FAA* compared with baselines.

---

[2]https://cloud.google.com/vision/docs/drag-and-drop

[3]See Section 7 for more information on the recruitment of volunteers and the evaluation procedure.

[4]https://www.reverseimagesearch.com/

## 7    ETHICS STATEMENT

This paper designs a novel approach to enhance the transferability of adversarial examples. While this approach is easy-to-implement and seems harmful, it is believed that the benefits of publishing *FAA* outweigh the potential harms. It is better to expose the blind spots of DNNs as soon as feasible because doing so can alert deployers to be aware of potential threats and greatly encourage AI community to design corresponding defense strategies.

For the human assessment process (Section 5), we generated adversarial examples and collected the predictions of different applications on these adversarial examples. Throughout the entire process, all communication with the volunteer, including recruitment, was conducted anonymously. Similarly, we do not knew the personal information regarding to the volunteer. The volunteer were unaware of our specific objectives, ensuring that there was no interest between the volunteer and us. The volunteer came from a certain university and possessed normal discernment abilities. During the rating process, the volunteer were unaware of whether a given sample was an adversarial example or the attack method used to generate it. Therefore, there was no bias towards any particular type of attack method from the volunteer. Overall, the evaluation process was relatively fair.

## 8    REPRODUCIBILITY

To ensure that researchers can replicate our experiments, we provide detailed descriptions of methods, experimental setups, and code implementations. The source codes are available in the supplementary materials for review. Additionally, a detailed readme file is provided to assist researchers in comprehending and utilizing the code effectively. We also supply dependency information to guarantee smooth execution of the code. After the paper is accepted, we will make all source codes publicly accessible on GitHub.

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

## A    MORE RESULTS ABOUT LAUNCH EXPERIMENT

We here provide more results to validate our point. Figure 6 shows the results in MobileNetV2 and EfficientNet. As can be seen, transferable adversarial examples usually enjoy small variance, *i.e.*, falling in flat regions.

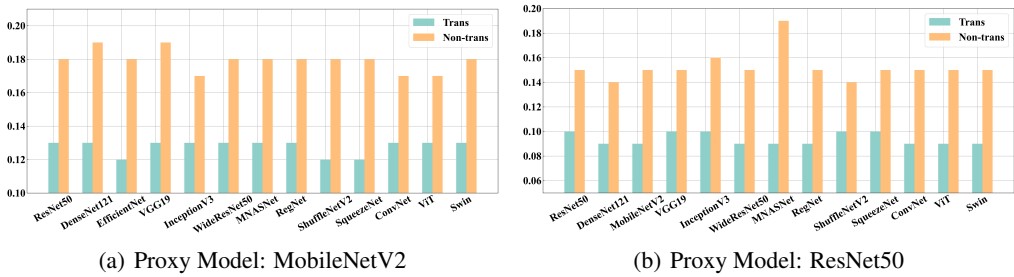

(a) Proxy Model: MobileNetV2                    (b) Proxy Model: ResNet50

Figure 6: The variance of model prediction for transferable and non-transferable adversarial examples when deploying MobileNetV2 and EfficientNet as proxy model.

## B    PROBLEM IN SOLVING BI-LEVEL TASK

The task of solving bi-level problems, also known as min-max problems, is widely recognized to be challenging. For more details, please refer to (Boyd & Vandenberghe, 2004). We here provide an example to show the difficulty of gradient descent algorithm in solving bi-level problems.

Let $f(\theta, \delta) = min_\theta max_\delta\ \theta\delta$, where $\theta$ and $\delta$ are defined on the interval $[-1, 1]$. For the inner problem, we see $max_\delta \theta\delta = |\theta|$, and the optimal solution is $\theta = 0$. Consider the case $\theta = 0$. In this case, the inner optimal solution $\delta$ can be arbitrary values between -1 to 1. Without loss of generality, we set $\delta = 1$ and we observe $\nabla_\theta f(\theta, \delta) = f(0, 1) = 1 \neq 0$. However, $\theta = 0$ already is the optimal solution for the bi-level task. To sum up, we see gradient descent algorithm may cause suboptimal solutions for bi-level problems.

## C    THE CONNECTIONS BETWEEN THE FLATNESS-AWARE ITEM AND THE TRANSFERABILITY

The effectiveness of adversarial examples against target models is referred to as their transferability. In this paper, we break down transferability into two factors: *the local effectiveness term*, which measures the loss on the proxy model, along with *the transfer-related loss term*, which quantifies the change in the loss of adversarial examples when transferring from the proxy model to the target model. A higher local effectiveness term and a lower transfer-related loss signify a stronger transferability of adversarial examples. In the following, we demonstrate that, minimizing the flatness-aware item in Equation 4 can be seen as a way to minimize the transfer-based loss to a certain extent, thus validating the effectiveness of Equation 4.

Assume the dataset of interest follows distribution $p(x, y)$ and there exists a ground-truth model $F_g(\cdot)$ that fits $p(x, y)$ perfectly. We employ cross-entropy loss as the loss function. We first consider $F_g(\cdot)$ to be the target model. By employing Taylor expansion on $\mathcal{L}(F_g(x), y) - \mathcal{L}(F(x), y)$, we derive the following expression:

$$
\begin{aligned}
&\mathcal{L}(F_g(x), y) - \mathcal{L}(F(x), y) \\
&= L(F_g(x + \delta), y) - \nabla L(F_g(x + \delta), y)^T\delta - L(F(x + \delta), y) + \nabla L(F(x + \delta), y)^T\delta.
\end{aligned}
\tag{9}
$$

Wherein, $L(F_g(x + \delta), y) \approx L(F(x + \delta), y)$ is our transfer-related loss term. We assume that our proxy model can fit $p(x, y)$ well. Therefore, due to $\mathcal{L}(F_g(x), y) - \mathcal{L}(F(x), y) \approx 0$, we wish $\nabla L(F_g(x + \delta), y)^T\delta - \nabla L(F(x + \delta), y)^T\delta \approx 0$ so as to make $L(F_g(x + \delta), y) \approx L(F(x + \delta), y)$, *i.e.*, small transfer-related loss term. For simplicity of notation, in the following analysis, we omit

$y$. We consider minimizing $||\nabla L(F_g(x+\delta), y)^T\delta - \nabla L(F(x+\delta), y)^T||_2^2$ over the distribution of interest:

$$\int p(x)||(\nabla log p(x+\delta) - \nabla log F(x+\delta))^T\delta||_2^2 dx$$

$$\leq \int p(x)||\delta||_2^2||\nabla log p(x+\delta) - \nabla log F(x+\delta)||_2^2 dx \quad (by \quad ||ab|| \leq ||a||||b||)$$

$$= \int p(x)||\delta||_2^2(||\nabla log p(x+\delta)||_2^2 + ||\nabla log F(x+\delta)||_2^2) dx$$

$$- 2\int p(x)||\delta||_2^2 \nabla log p(x+\delta)^T \nabla log F(x+\delta) dx. \tag{10}$$

For the second integral $\int p(x)||\delta||_2^2 \nabla log p(x+\delta)^T \nabla log F(x+\delta) dx$ in the above equation, we have:

$$\int p(x)||\delta||_2^2 \nabla log p(x+\delta)^T \nabla log F(x+\delta) dx$$

$$= \int \frac{p(x)}{p(x+\delta)}||\delta||_2^2 \nabla p(x+\delta)^T \nabla log F(x+\delta) dx$$

$$\geq \int ||\delta||_2^2 \nabla p(x+\delta)^T \nabla log F(x+\delta) dx \quad (use \quad p(x) \geq p(x+\delta)) \tag{11}$$

$$= ||\delta||_2^2 p(x+\delta)^T \nabla log F(x+\delta)|_{-\infty}^{+\infty} - \int ||\delta||_2^2 p(x+\delta)^T \nabla^2 log F(x+\delta) dx$$

$$= -\int ||\delta||_2^2 p(x+\delta)^T \nabla^2 log F(x+\delta) dx.$$

In the third line of Equation 11, we use $p(x) \geq p(x+\delta)$. This is because that, $\delta$ is crafted for attacking the proxy model. The proxy model makes worse predictions for $x + \delta$ than $x$. Furthermore, the similarity between the proxy and target model makes it work in the target model. In the fourth line of Equation 11, we suggest $p(+\infty) = p(-\infty) = 0$, which is natural in the distribution of interest.

Combining Equation 10 and 11, we have:

$$\int p(x)||(\nabla log p(x+\delta) - \nabla log F(x+\delta))^T\delta||_2^2 dx$$

$$\leq \int p(x)||\delta||_2^2(||\nabla log p(x+\delta)||_2^2 + ||\nabla log F(x+\delta)||_2^2) dx + 2\int ||\delta||_2^2 p(x+\delta)\nabla^2 log F(x+\delta) dx$$

$$\leq \int p(x)||\delta||_2^2(||\nabla log p(x+\delta)||_2^2 + ||\nabla log F(x+\delta)||_2^2) dx + 2\int ||\delta||_2^2 p(x)^T ||\nabla^2 log F(x+\delta)|| dx. \tag{12}$$

In the second line of Equation 11, we use $p(x) \leq p(x+\delta)$. We see that decreasing $||\nabla log p(x+\delta)||_2^2$, $||\nabla log F(x+\delta)||_2^2$, and $||\nabla^2 log F(x+\delta)||$ can enhance the transferability of crafted adversarial examples. Now we back to our flatness-aware regularization item $||\nabla \mathcal{L}(F(x+\delta+\Delta), y)||_2, \Delta \sim U(-b, b)$. We find the flatness-aware item in fact minimizes the above three items. Notice that the flatness-aware item punishes the norm of gradients of samples around $x + \delta$. Therefore, this induces $||\nabla log F(x+\delta)||_2^2$ and $||\nabla^2 log F(x+\delta)||$ to be 0. Moreover, if the proxy and the target model are similar, the gradients of the proxy and target model also present high similarity. In practice, the target model may not perfectly fit the distribution of interest but can fit it well. If adversarial examples are effective against the ground-truth model, they are likely to be effective against the target model as well. In short, we grasp why minimizing flatness-aware item can be seen as a way to minimize the transfer-based loss.

## D APPROXIMATION ERROR OF EQUATION 8

We here evaluate the approximation error of our method (Equation 8). Specifically, we compute the ground-truth Hessian matrix and derive the ground-truth values of $H_{x+\delta+\Delta_i} \frac{\nabla \mathcal{L}(F(x+\delta+\Delta_i), y)}{||\nabla \mathcal{L}(F(x+\delta+\Delta_i), y)||_2}$. Table 5 reports the averaged maximum absolute distance between the ground-truth values and the approximated values obtained using Equation 8 over 100 samples. The results demonstrate that the approximation error diminishes as the value of $k$ decreases.

Table 5: Approximation error of our method with varying $k$.

| k | 0.001 | 0.01 | 0.1 | 1.0 |
|---|---|---|---|---|
| Approximation Error | 0.001 | 0.0042 | 0.0092 | 0.0114 |

# E SUPPLEMENTARY EXPERIMENTS

## E.1 ATTACK RESULTS ON OTHER DEFENSES

We here evaluate the attack performance of *FAA* against other defenses including R&P (Xie et al., 2017), NIPS-R3 [5], FD (Liu et al., 2018), ComDefend (Jia et al., 2018), RS (Jia et al., 2020), NRP (Naseer et al., 2020). These defense mechanisms deviate significantly from adversarial training as they rely on data processing techniques to purify adversarial examples into normal ones. The detailed settings follow Long et al. (2022). Table 6 reports the attack performance of VT, SSA, RAP, and our method against these defenses. *FAA* still consistently surpasses baselines by a large margin.

Table 6: The attack success rates (%) of attacks against various defenses. We use ResNet50 as the proxy model.

| Attack | R&P | NIPS-R3 | FD | ComDefend | RS | NRP |
|---|---|---|---|---|---|---|
| VT | 64.53 | 70.69 | 73.4 | 87.33 | 52.14 | 43.97 |
| SSA | 72.10 | 75.32 | 76.06 | 91.27 | 61.86 | 52.91 |
| RAP | 93.15 | 92.44 | 92.58 | 95.59 | 76.09 | 76.28 |
| Ours | **95.81** | **97.44** | **94.02** | **97.64** | **85.54** | **83.47** |

## E.2 ATTENTION VISUALIZATION FOR TARGETED ATTACKS

Figure 7 shows more visualizations of attention maps. As can be seen, the adversarial examples produced by VR and SSA only can slightly decrease the attention of the target model to entities of images, while *FAA* can significantly distract the attention of the target model from the entities to trivial regions.

## E.3 IMPACT OF HYPERPARAMETERS

*FAA* involves four hyperparameters that can significantly impact the attack performance:

- $\lambda$ is a balance factor between the loss of resultant adversarial examples and the flatness of the region around the adversarial ones. The bigger $\lambda$ attaches more attention to promoting adversarial examples toward flat regions.
- $b$ suggests how wide the region around adversarial examples is desired to be flat.
- When comes to the implementation of *FAA*, Hessian matrix is approximately evaluated and a inappropriate $k$ probably induces non-trivial errors.
- We extract $N$ samples around adversarial examples to estimate the flatness of the region around the adversarial ones. Generally, the estimation accuracy raises with increasing N.

**The impact of** $\lambda$**.** Figure 8(a) illustrates the attack success rates of crafted adversarial examples by *FAA* with varying $\lambda$ against target models. We see that as $\lambda$ increases, the attack effectiveness of *FAA* presents the tendency to climb initially and decline thereafter. If $\lambda$ is small like $\lambda = 0.1$, most attention of Equation 4 is put into optimizing $L$ while ignoring the flatness of the region around crafted adversarial examples, *i.e.*, degrading into vanilla transfer-based attacks and causing that the adversarial ones are more likely to be trapped by model-specified sharp regions. As a remedy, properly increasing $\lambda$ can grab part of the attention of Equation 4 to focus on the flatness of the surrounding region, so that the resultant adversarial examples have more chance to evade the model-specified

---

[5]https://github.com/anlthms/nips-2017/blob/master/poster/defense.pdf

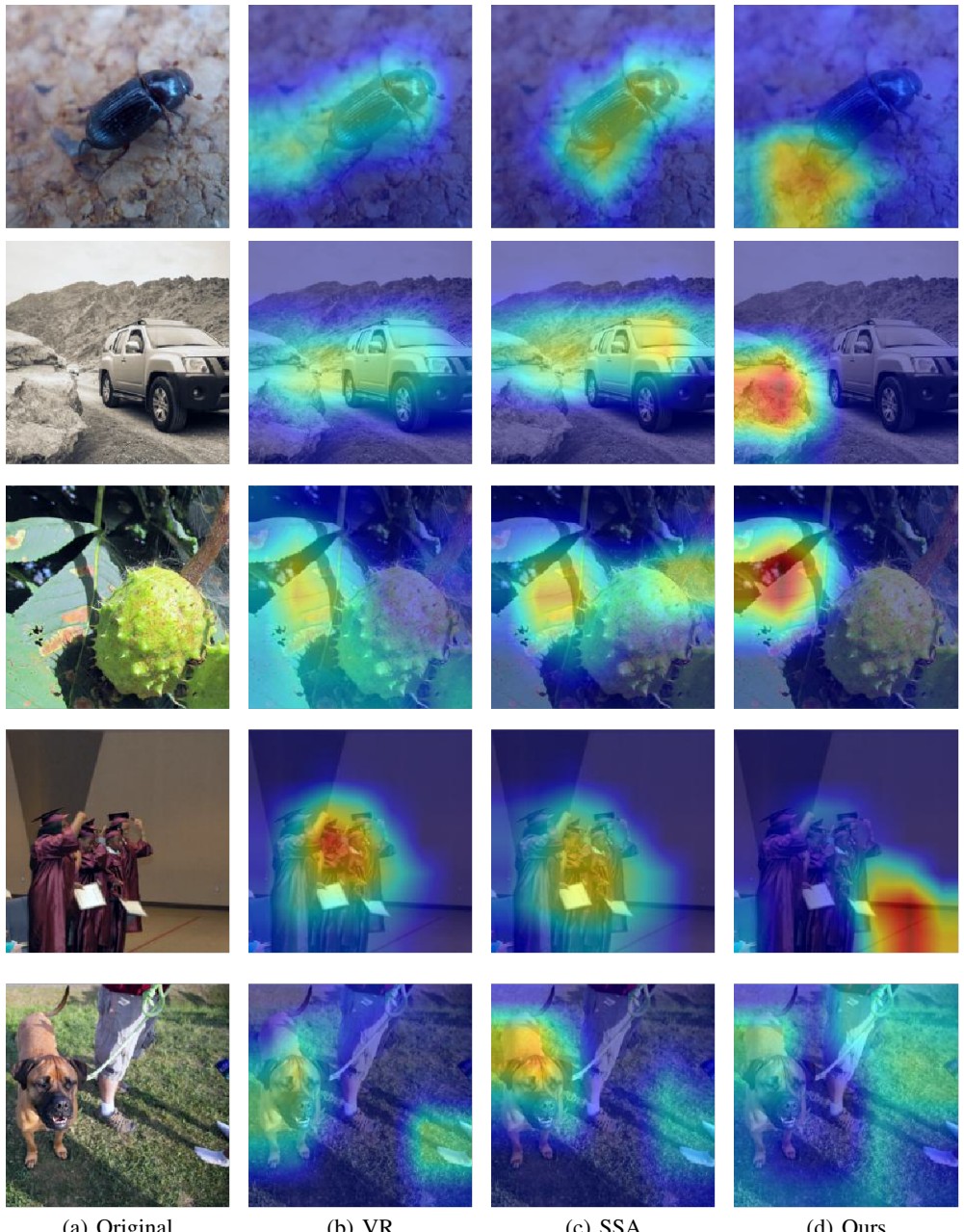

|        |        |         |          |
|:------:|:------:|:-------:|:--------:|
| (a) Original | (b) VR | (c) SSA | (d) Ours |

Figure 7: We conduct targeted attacks and visualize attention maps of the target model to the resultant adversarial images.

sharp regions and are more transferable. However, it should be stressed that too bigger $\lambda$ induces Equation 4 to only focus on the flatness and de-emphasize whether the region poses a threat to the proxy model, leading to a reduction in the attack success rate. Therefore, carefully adjusting $\lambda$ is necessary.

**The impact of** $b$. We examine the attack effectiveness of *FAA* by changing the hyperparamter $b$ and Figure 8(b) shows the attack results over different proxy-target pairs. Overall, the attack performance of *FAA* steadily increases with increasing $b$. The reason for it is that, a small $b$ makes the extracted samples too close to the resultant adversarial examples and causes that the flatness of the region around the resultant adversarial examples cannot be effectively evaluated. Furthermore, increasing $b$

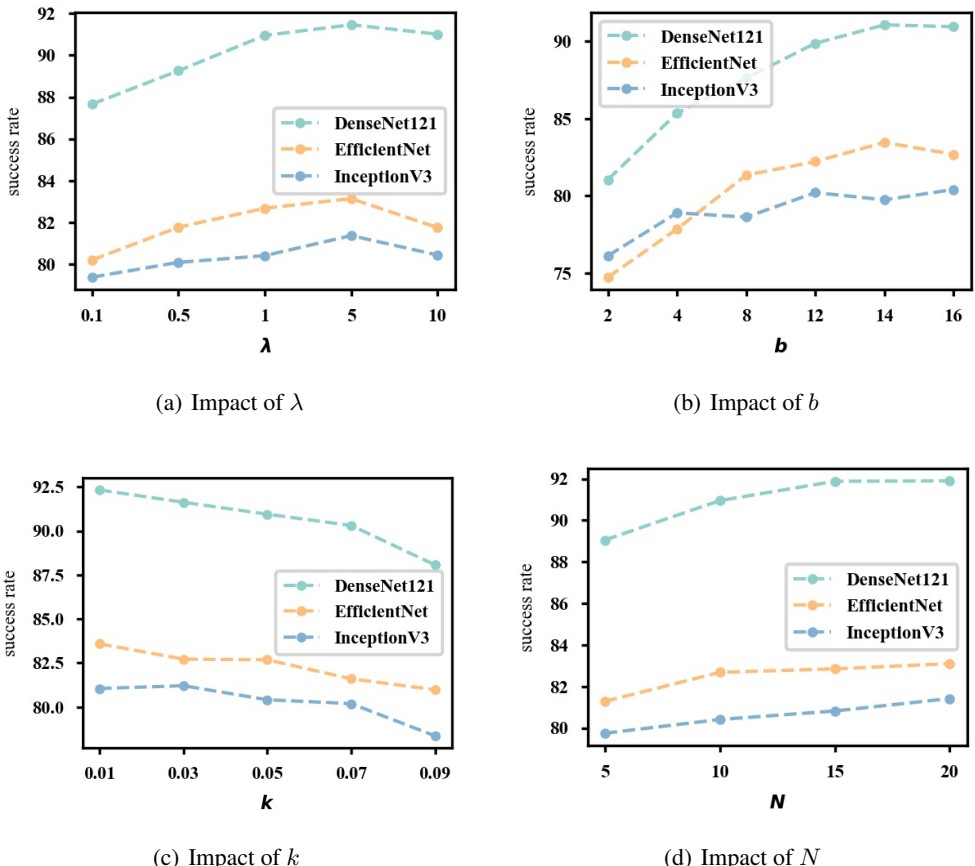

(a) Impact of $\lambda$

(b) Impact of $b$

(c) Impact of $k$

(d) Impact of $N$

Figure 8: The attack effectiveness of *FAA* with varying $\lambda \in \{0.1, 0.5, 1, 5, 10\}, b \in \{1, 2, 4, 8, 12, 16\}, k \in \{0.01, 0.03, 0.05, 0.07, 0.09\}, N \in \{5, 10, 15, 20\}$. The proxy model is ResNet50. We set $\epsilon = 8$.

alleviates the issue so as to boost the transferability of produced adversarial examples. Moreover, interestingly, we find that, when employing ResNet50 and EfficientNet, increasing $b$ from 14 to 16 slightly hurts the transferability of produced adversarial examples, and this is probably attributed to the extracted samples being a little far from the crafted adversarial examples.

**The impact of $k$.** Figure 8(c) shows the influence of the hyperparameter $k$ to the performance of *FAA* and we observe that increasing $k$ weakens the attack effectiveness of *FAA*. In fact, as shown in Section 3.2, Hessian matrix is approximately replaced by gradient difference, *i.e.*, Equation 8, and the feasibility of the approximation solution depends on a small $k$ to omit the error of approximation induced by Taylor expansion. Hence, it is intuitive why a bigger $k$ incurs degradation on the attack success rates of *FAA*.

**The impact of $N$.** Figure 8(d) shows the attack performance of *FAA* with different $N$. Intuitively, by sampling more instances from the region around the resultant adversarial examples, *i.e.*, increasing $N$, we can make a more accurate estimation of the flatness of the region, which in turn can decrease estimation errors and then strengthen the transferability of generated adversarial examples. As expected, Figure 8(d) validates this point that increasing N promotes the transferability of produced adversarial examples.

### E.4 ATTACKS WITH $L_2$ CONSTRAINT

Table 7 reports the attack results with $L_2$-norm constraint. The attack effectiveness of *FAA* still outperforms baselines by a large margin.

Table 7: Thr attack results with $L_2$-norm constraint. We use ResNet50 as the proxy model.

| Attack | DenseNet121 | EfficientNet | InceptionV3 |
|---|---|---|---|
| VR (Wang & He, 2021) | 81.88 | 79.84 | 73.49 |
| SSA (Long et al., 2022) | 83.96 | 83.02 | 77.65 |
| Ours | 87.48 | 85.68 | 80.05 |

## E.5 ATTACKS WITH VARYING ITERATION

We here investigate the impact of iterations on attack performance. Table 8 reports the performance of different attacks with varying numbers of iterations. It can be observed that there is negligable change in attack performance when the number of iterations is increased from 10 to 20. The results suggest that the attack methods achieve convergence with 10 iterations.

Table 8: The attack effectiveness of different attacks with varying iterations. We use ResNet50 as the proxy model.

| Attack | ResNet50 | DenseNet121 | EfficientNet |
|---|---|---|---|
| VT (10 iter) | 100.00 | 88.76 | 81.51 |
| VT (20 iter) | 100.00 | 89.62 | 81.63 |
| SSA (10 iter) | 100.00 | 95.29 | 90.72 |
| SSA (20 iter) | 100.00 | 95.51 | 91.56 |
| Ours (10 iter) | 99.80 | 99.08 | 99.25 |
| Ours (20 iter) | 99.85 | 99.69 | 99.52 |

## E.6 COMPARISON TO TRANSFORMER-BASED ATTACKS

Recent certain attacks (Wei et al., 2021; Naseer et al., 2021) exploit the unique characteristics of transformer-like networks to conduct transfer-based attacks on similar networks. Intuitively, these attacks achieve better attack performance against transformer-like networks due to the higher similarity of the proxy model to the target model. We evaluate the attack performance of such attacks and *FAA* and the results are reported in Table 9. Notably, the attack success rates of these attacks are less than 80%. In contrast, *FAA* shows impressive attack performance, surpassing the baseline by achieving a remarkable 90.98% attack success rate against Swin.

Table 9: The attack success rates (%) of different attacks against Swin. MASR and RE use ViT as the proxy model. *FAA* employs ResNet50 as the proxy model.

| Attack | MASR Wei et al. (2021) | PGD+RE Naseer et al. (2021) | Ours |
|---|---|---|---|
| ASR | 46.10 | 77.40 | 90.98 |

## E.7 VALIDATION OF GENERALIZATION OF *FAA*

Table 10 validates that *FAA* is a generic method. By combining *FAA*, the attack effectiveness of the three methods can be notably enhanced.

## E.8 FLATNESS COMPARISON

We compare the flatness of adversarial examples generated by *FAA* with that of RAP, following the setting of the figure 6 of the RAP original paper Qin et al. (2022). As shown in Figure 9, *FAA* produces flatter adversarial examples than *FAA*, providing an explanation for the better effectiveness of *FAA*. Regarding the inferior flatness of RAP's adversarial examples, there are two reasons. The first one lies in the difficulty of solving bi-level optimization (See Appendix B). Secondly, as mentioned in Section 1, the optimal transformation technique or combination thereof remains unclear.

Table 10: The generalization validation of *FAA*. We use $\epsilon$ of 8.

| Attack | DenseNet121 | EfficientNet | InceptionV3 |
|---|---|---|---|
| PGD (Madry et al., 2018) | 29.88 | 24.76 | 22.14 |
| PGD+Ours | 91.61 | 83.14 | 80.72 |
| DI (Xie et al., 2019) | 40.25 | 36.02 | 33.98 |
| DI+Ours | 92.18 | 83.18 | 80.82 |
| SGM (Wu et al., 2020) | 57.36 | 52.72 | 49.36 |
| SGM+Ours | 92.29 | 83.19 | 81.89 |

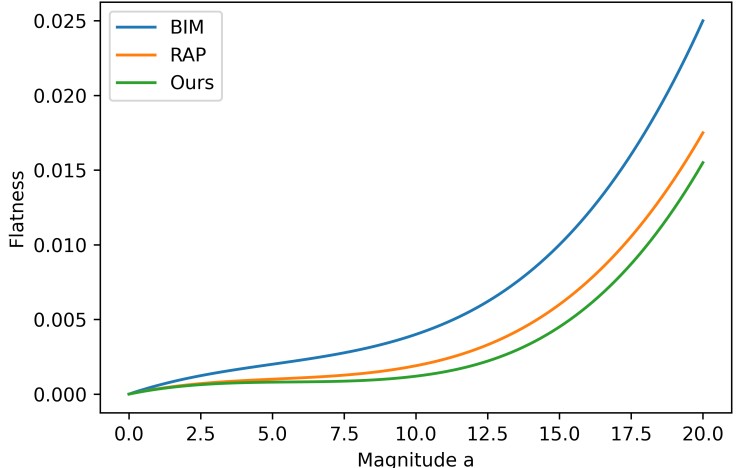

Figure 9: The visualization of flatness. A lower value represents better flatness.

### E.9 COMPARISON OF ATTACKS ON REAL-WORLD CV APPLICATION

Table 11 reports the attack performance of VT, SSA, and *FAA* on classification and object detection in Google MLaaS platforms. The results demonstrate the superiority of *FAA* over the baselines.

Table 11: The effectiveness of attacks against real-world applications. We report the total number of samples scored by 1, 2, and 3 for each attack.

| | Classification | Object Detection |
|---|---|---|
| VT | 56 | 32 |
| SSA | 71 | 48 |
| Our | **92** | **76** |

## F SCORE IMPLICATION

We begin by gathering predictions from Google MLaaS platforms for both label detection and object detection over crafted adversarial examples. We assign scores to these predictions along five discrete levels, with scores of 1 through 5 indicating the degree of accuracy: totally wrong, slightly incorrect, strange but not incorrect, relatively reasonable, and precise. A score of 1 suggests that the images do not contain the objects predicted. A score of 2 indicates minor errors in the predictions, such as identifying flowers instead of trees in a tree image. A score of 3 suggests that the main object in the image is not correctly recognized, for example, predicting stones, roads, or tires for a car image. A score of 4 denotes accurately identifying the general type of the main object in the images, while a score of 5 indicates that the system can fully and correctly identify the main object in the image. Next, a volunteer (See Section 7 for more information on the recruitment of volunteers and the evaluation procedure) is asked to rate the consistency between the predictions and the images based on these

scores For search engines, a similar evaluation procedure is followed, where we assess the similarity between the retrieved images and their corresponding original images.

## G A VISUALIZATION OF DIFFERENT ATTACKS AGAINST GOOGLE SERVICE AND SEARCH ENGINES

Figure 10 shows a visualization of different attacks against Google Service, and we use the format {attack method}-{task} to denote the attack used to craft adversarial examples and the corresponding test task.

For image classification, consistent with our intuition, the original image is predicted into plant-related categories. However, the returned predictions for the adversarial examples produced via VR and SSA are close to the ground-truth label of the original image. Therefore, the adversarial examples cannot be deemed to be threats against Google Cloud Vision. In contrast, the adversarial examples generated by our method indeed mislead Google Cloud Vision, where the predictions with the highest confidence are bird, water, and beak, and these predictions are fairly irrelevant to the original images.

For object detection, we can obtain similar conclusions. The original image is correctly detected. The adversarial examples produced by VR seem to fool Google Cloud Vision to some extent, while SSA can fully mislead Google Cloud Vision. Also, our adversarial examples trick Google Cloud Vision with higher confidence than VR.

Figure 11 visualizes an example of *FAA* against four state-of-the-art search engines. We observe that search engines fetch high-quality and similar images for normal samples. However, when we input the generated adversarial examples, the quality of retrieved images noticeably deteriorates, particularly in the case of Baidu.

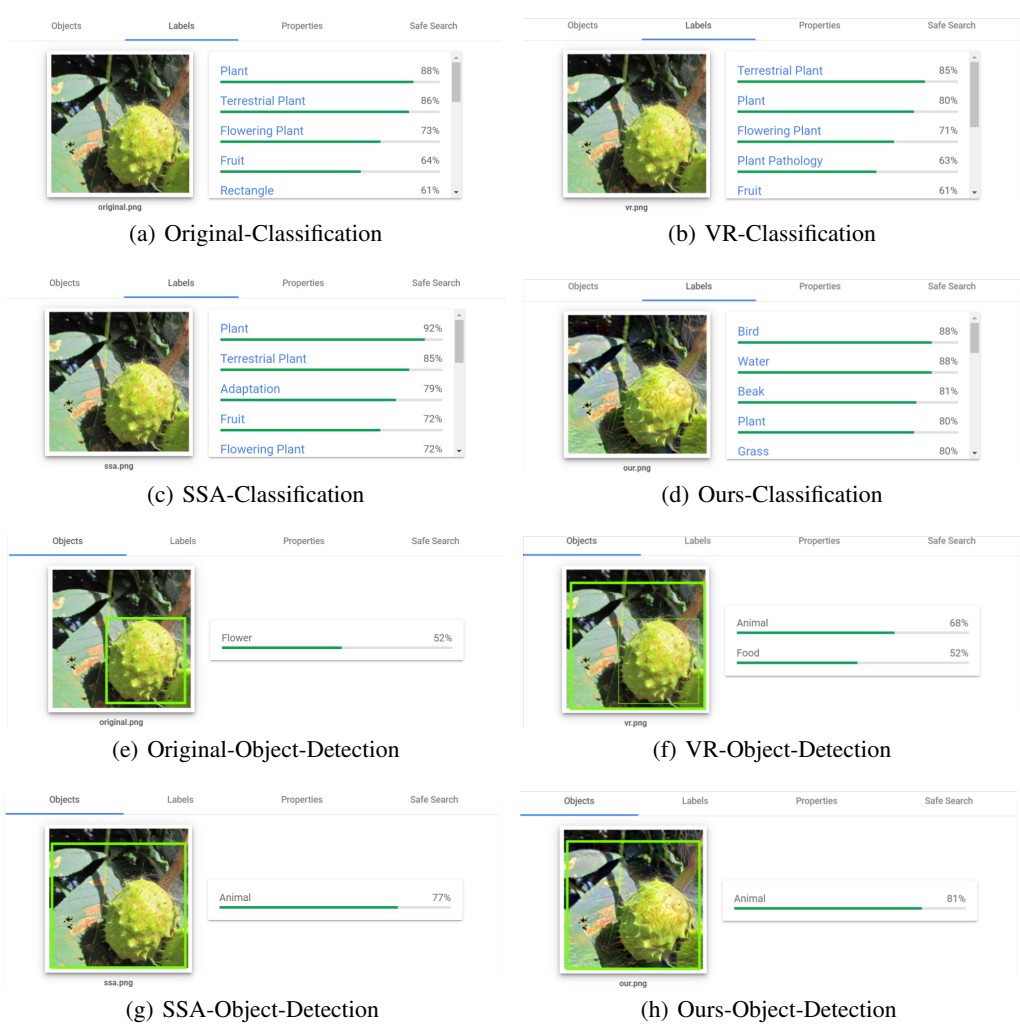

Figure 10: The attack results on Google Cloud Vision including image classification and object detection tasks. We do not know any knowledge of it and the proxy model is ResNet50.

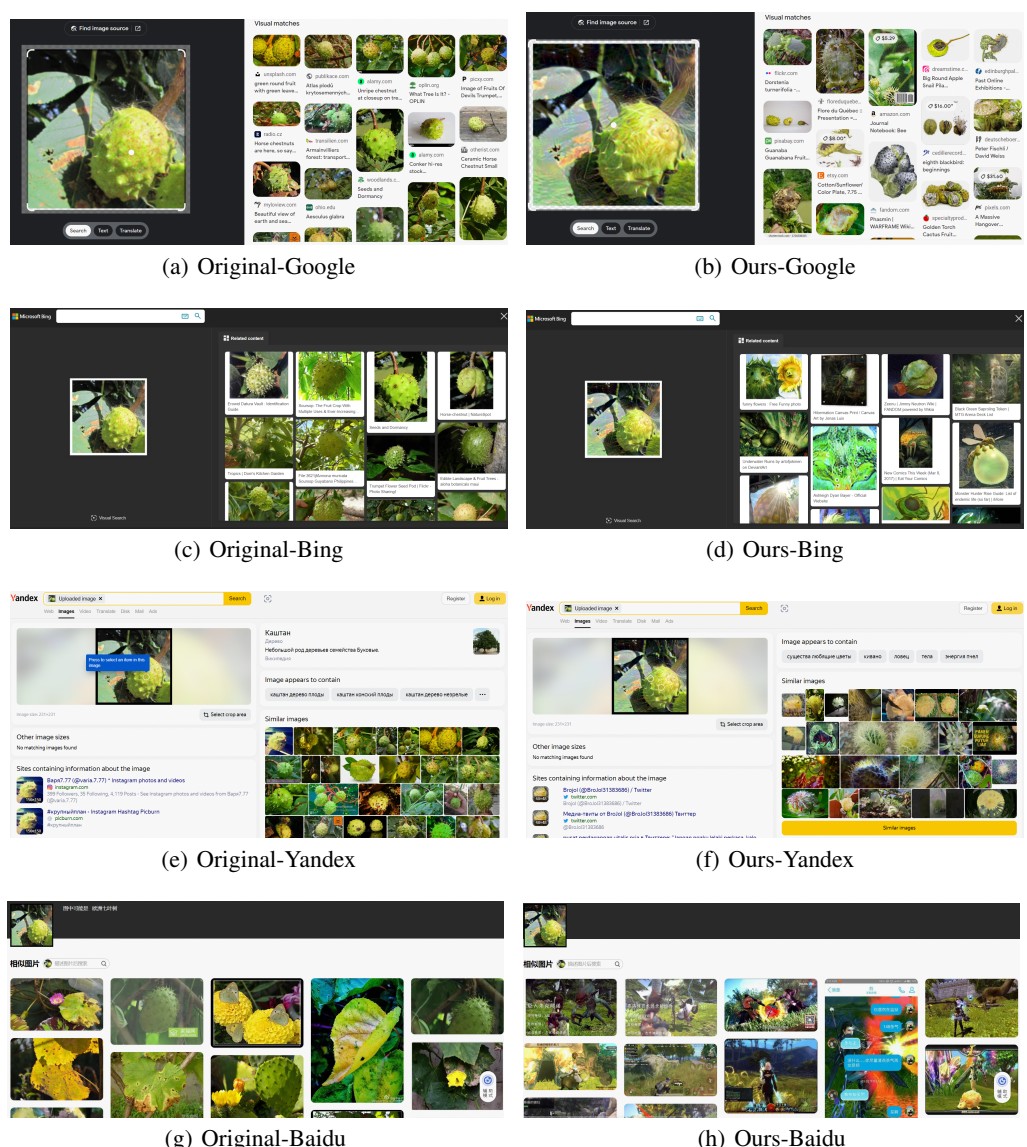

Figure 11: An example for attacking four state-of-the-art search engines.

