# OpenReview forum: "Flatness-aware Adversarial Attack"
_ICLR.cc/2024/Conference — ICLR 2024 Conference Withdrawn Submission_

### Official Review · Reviewer_fujR · 2023-10-18

**Soundness:** 3 good
**Presentation:** 3 good
**Contribution:** 1 poor
**Rating:** 3
**Confidence:** 5

**Summary:**

In this work, the authors propose flatness-aware adversarial attack (FAA) to improve the adversarial transferability. In particular, FAA adopts a regularizer for the gradient to make the generated adversarial example located in a flat local optimum. To avoid the Hessian matrix calculation, they utilize Taylor expansion to approximate the Hessian matrix. Experiments on ImageNet dataset show the effectiveness of FAA.

**Strengths:**

1. The paper is well-written and easy to follow.

2. The authors have conducted several experiments to validate the effectiveness of the proposed method.

**Weaknesses:**

1. The launch experiment is not solid enough. In my opinion, the experiments can only conclude that transferable adversarial examples might be in flatter local optima. It is a necessary but not sufficient condition that adversarial examples in flatter local optima are more transferable.

2. It is not a new method that connects flat regions and input regularization methods [1]. Eq. (4) is similar to the Eq. (3) in [1], making the motivation rather limited. The main difference is how to approximate the Hessian matrix. Tylor expansion is not a novel way for such an approximation.

3. I am curious why did the author adopt the number of iterations $T=20$, since existing works mainly adopt $T=10$. Increasing the number of iterations might result in overfitting, which degrades the baselines' performance.

4. It is expected to see more momentum-based baselines, such as [1], [2].

5. Why does Table 2 miss the baseline RAP? It should be a significant baseline since it is the first paper that locates the adversarial examples in flat local optima for better transferability.


[1] Ge et al. Boosting Adversarial Transferability by Achieving Flat Local Maxima. arXiv Preprint arXiv: 2306.05225, 2023.

[2] Zhang et al. Improving the Transferability of Adversarial Samples by Path-Augmented Method. CVPR 2023.

**Questions:**

See weakness

---

> ### Author Response · Authors · 2023-11-12
> **Response (1/2)**
>
> Thank you for your efforts on our paper and your valuable suggestions. We answer your questions point-by-point as follows:
>
> **Question 1:**
> The launch experiment is not solid enough.
>
> **Response:**
> We would like to clarify some misunderstandings concerning our launch experiment.
> Our launch experiment aims to show that adversarial examples residing in flat regions enjoy better transferability (i.e., the necessary condition), instead of demonstrating the equivalence between the transferability of adversarial samples and flat regions.
> As shown by you, the results support our objective (the necessity).
>
> In fact, this paper delves into the transferability problem, with the hope of augmenting the transferability for the effectiveness of black-box attacks.
> In Section 1, drawing inspiration from input regularization methods, we hypothesize that adversarial examples located in flat regions enjoy better transferability.
> This conjecture is preliminarily supported by visualizations of the loss landscape (Figure 1).
> To further substantiate this hypothesis, we conduct the launch experiment.
> Leveraging the empirical evidence gathered, we are led to the conviction that adversarial examples in flat regions indeed possess superior transferability.
> Consequently, we propose our method, FAA, to improve the transferability of adversarial examples.
> By the way, Appendix C also offers a theoretical demonstration to support this hypothesis.
>
>
> **Question 2:**
> The novelty of this paper.
>
> **Response:**
> We carefully check [1] and think it is enlightening. However, this paper distinguishes itself from [1] by several key parts, constituting our contributions. The significance of transferability research to our community is two-fold: first, it enhances the understanding of transferability, which has implications for various related fields; second, it improves the effectiveness of black-box attacks, with potential implications for the development of defenses and increasing awareness of vulnerabilities in neural networks. Herein, we expound on how our work provides new insights into these two aspects.
>
> (I) Our motivation varies. Existing literature, including [1,2], analogizes the transferability of adversarial examples to models' generalization ability, inspiring exploration of techniques that enhance generalization to boost transferability, such as SAM-like optimization formulation [2]. Unlike these, to our best knowledge, we are the first to observe the potential associations between input regularization methods and flat regions (paragraph 1 in Intro). By clarifying how input regularization methods enable adversarial example convergence towards flat regions, we illuminate their limitations (paragraph 3 in Intro). This insight motivates the design of FAA (paragraph 3 in Intro). Thus, this paper deepens our understanding of existing input regularization methods by rethinking them from a unique perspective.
>
> (II) On the technical pathway, despite the similarity between Eq. 3 in [1] and Eq. 4 in our paper, their underlying technical trajectories differ fundamentally. They penalize maximum norms while we penalize gradients of samples around $x+\delta$. This subtle variation can indeed lead to significant differences. For instance, L1 and L2 regularizations appear highly similar in form, but their underlying mechanisms and effects are substantially different (L1 regularization induces sparsity). Specifically, from the perspective of conjugate functions [3], loosely speaking, Eq.3 in [1] would be a relaxation of Eq. 4 in our paper. In other words, our penalty term provides a tighter bound for flat regions, resulting in the superior performance of our method.
>
> (III) Our theoretical contribution is pioneering (Appendix C). To the best of our knowledge, we are the first to provide theoretical proof validating the relationship between transferability and flat regions. While [2] speculates on this relationship based on intuition, [1] follows suit. Their results are empirical, whereas ours possess theoretical underpinnings, positioning our paper in parallel rather than merely building upon (or overlapping) their work.
>
> (IV) The effectiveness of our method is significant. FAA is the first to achieve a success rate of over 90\% in attacking nearly all mainstream models, spanning CNNs and transformers, with or without defenses. Our attacks on real-world systems further demonstrate its effectiveness in the physical world. As evidenced in responses to questions 4 and 5, our attack success rate surpasses baselines by a clear margin. For practitioners and real-world applications, the primary concern lies in effectiveness rather than other factors. From this standpoint, our attacks hold immense value for industrial and DNN applications.
>
> In summary, these four significant parts constitute the contributions that set this paper apart from others, paving the way for advancements in this field.

---

> > ### Author Response · Authors · 2023-11-12
> > **Response (2/2)**
> >
> > **Question 3:**
> > I am curious why did the author adopt the number of iterations T=20, since existing works mainly adopt T=10. Increasing the number of iterations might result in overfitting, which degrades the baselines' performance.
> >
> >
> > **Response:**
> >
> > Our approximation involves Hessian matrix, which is generally believed to accelerate convergence.
> > To eliminate this confounding factor, we increase the number of iterations to ensure the convergence of all attacks.
> > Furthermore, we take into account the issue of overfitting and include experiments with varying iterations in Appendix, as per Table 8 in the original paper.
> > Upon examination of the results, the increased iterations did not significantly affect the attack performance (less than 1%).
> >
> >
> >
> > **Question 4 \& 5:**
> > It is expected to see more momentum-based baselines, such as [1], [2].
> > &
> > Why does Table 2 miss the baseline RAP? It should be a significant baseline since it is the first paper that locates the adversarial examples in flat local optima for better transferability.
> >
> > **Response:**
> > We adopt your suggestion and include the baselines you mentioned for comparison. Moreover, we add the attack results of RAP in Table 2.
> >
> > All new attack results can be found in the revised paper, which will be uploaded following the consolidation of feedback from all reviewers prior to the conclusion of the review period (we have inquired about other reviewers whether there exist new suggestions).
> > We here provide part of the attack results shown in the table below.
> > Simply speaking, we see the effectiveness of our method still consistently surpasses baselines by a large margin.
> >
> > | Attack | ConvNet | ViT    | Swin   | AdvIncV3 | EnsAdvIncResV2 | L2-0.03 | L2-5    |
> > |:------:|:-------:|:------:|:------:|:--------:|:--------------:|:-------:|:-------:|
> > | PAM [4]    | 80.63   | 58.38  | 55.75  | 59.28    | 46.05          | 79.79   | 78.34   |
> > | RAP [2]    | 90.80   | 62.62  | 60.77  | 60.17    | 51.02          | 81.72   | 76.05   |
> > | PGN [1]    | 92.15   | 67.83  | 66.38  | 63.02    | 55.83          | 85.34   | 80.61   |
> > | Ours   | 94.34   | 93.52  | 90.98  | 74.57    | 72.81          | 96.20   | 87.24   |
> >
> > Reference:
> >
> > [1] Boosting Adversarial Transferability by Achieving Flat Local Maxima, arXiv 2023.
> >
> > [2] Boosting the Transferability of Adversarial Attacks with Reverse Adversarial Perturbation, NIPS 2022.
> >
> > [3] SAM as an Optimal Relaxation of Bayes, ICLR 2023.
> >
> > [4] Improving the Transferability of Adversarial Samples by Path-Augmented Method, CVPR 2023.

---

> > > ### Author Response · Authors · 2023-11-14
> > > **Discussion Inquiry**
> > >
> > > Dear Reviewer,
> > >
> > > We thank you for the precious review time and valuable comments. We have provided responses to your question and the weakness you mentioned. We hope this can address your concerns.
> > >
> > > We hope to further discuss with you whether or not your concerns have been addressed appropriately. Please let us know if you have additional questions or comments. We look forward to hearing from you soon.
> > >
> > > Best regards,
> > >
> > > Authors

---

> > > > ### Comment · Reviewer_fujR · 2023-11-15
> > > > **Thanks for the reply**
> > > >
> > > > Thank the authors for the reply. I have read the rebuttal and the other reviews. However, I still think my concerns have not been addressed.
> > > >
> > > > + The motivation and approach are similar to [1]. Also, flat local maxima resulting in better transferability is not a new idea [2].
> > > >
> > > > + I have read the paper again and am confused if Eq. (7) holds. Eq. (6) is a Taylor expansion, ensuring the equivalence of the function's value within a limited vicinity. Nevertheless, this does not inherently guarantee a corresponding equivalence in their gradients. Consequently, it appears that Equation (7) may not be a direct derivative of Equation (6).
> > > >
> > > > With the concerns about the novelty and approach, I tend to maintain my score currently.
> > > >
> > > > [1] Boosting Adversarial Transferability by Achieving Flat Local Maxima, arXiv 2023.
> > > >
> > > > [2] Boosting the Transferability of Adversarial Attacks with Reverse Adversarial Perturbation, NIPS 2022.

---

> > > > > ### Author Response · Authors · 2023-11-16
> > > > > **Response to novelty and effectiveness of Taylor expansion**
> > > > >
> > > > > Thanks for your reply.
> > > > >
> > > > > **Question 1: Novelty.**
> > > > >
> > > > > **Response:**
> > > > > We want to make further clarification on what is new in this paper.
> > > > >
> > > > > We unveil the relationship between input regularization methods and flat regions, shedding light on the underlying reasons for the effectiveness of input regularization methods. Our contribution lies in offering a more unified and deeper understanding of these methods.
> > > > >
> > > > > We present theoretical evidence demonstrating the better transferability of adversarial samples in flat regions. Theoretical proofs serve to validate the logical and mathematical soundness of an idea, thereby enhancing its credibility. Unless one considers theoretical proofs to be trivial.
> > > > >
> > > > > FAA is the first attack to achieve such a high attack success rate. If we can fulfill an idea better in another technique pathway, is this trivial? We don't think so.
> > > > >
> > > > >
> > > > > **Question 2: The effectiveness of Taylor expansion.**
> > > > >
> > > > > **Response:**
> > > > > The effectiveness of Taylor expansion is established within a small vicinity.
> > > > > This implies that the estimation error can be tolerated if we operate within a small vicinity.
> > > > > In Eq. 7, we only need to select a smaller value for $\phi$ (i.e., selecting $k$ in practice), which guarantees a small vicinity.
> > > > > Table 5 also reports the approximate errors.
> > > > > Moreover, the effectiveness of our attack method also supports the validity of this approximation.

---

> > > > > > ### Author Response · Authors · 2023-11-17
> > > > > > **Discussion Inquiry**
> > > > > >
> > > > > > Dear Reviewer,
> > > > > >
> > > > > > We are delighted to engage in further discussions regarding the concerns you have raised.
> > > > > >
> > > > > > We await your response and sincerely appreciate the efforts you have put into reviewing this paper.
> > > > > >
> > > > > > Best regards,
> > > > > >
> > > > > > Authors

---

### Official Review · Reviewer_tmEk · 2023-10-30

**Soundness:** 3 good
**Presentation:** 4 excellent
**Contribution:** 3 good
**Rating:** 5
**Confidence:** 4

**Summary:**

The paper presents an approach to improve the transferability of adversarial attacks using the properties of input signals: the more flat region on the loss curve they occupy, the more transferrable they are.

**Strengths:**

A paper seems to provide a very good method because:
- it provides almost 100% transferability for both normal (Table 1) and secured (Table 2) models (!) for untargeted attacks with a huge margin over other methods
- it beats other methods for targeted attacks (Table 3) - again with a significant margin
- it provides a solid reasoning based on observations, and the theory behind it
- and even the theory was adopted towards the fast computation cycle (to get rid of Hessian matrix computation)
- it was tested in real CV applications

**Weaknesses:**

Although the paper provide a lot of insights, there are still some (I hope minor and improvable) drawbacks:
- Page 4, the ref. to Mean Value Theorem: it'd be better to refer to some classic mathematical results (I guess they are discovered hundreds of years ago, not just 15)
- Page 16, Table 5: It's not clear what "Approximation Error" exactly means: is it the overall Hessian-based additive term, or Term1, or Term2 (Equation 8)?
- Page 19, Appendix E.8: "As shown in Figure 9, FAA produces flatter adversarial examples than FAA" -> "As shown in Figure 9, FAA produces flatter adversarial examples than RAP"?
- But my main concern is the (based on my assessment) theory-related inference on Pages 15 and 15 (Appendix C). Let me briefly provide it so the errors could be corrected:

First of all, I'm not sure that the Eq. (9) is correct. It uses a Taylor Expansion of a complex function (which in turn relies on the derivative of a complex function), so I think the correct way to do it: $f(g(x))=f(g(a))+\nabla g(a)\nabla f(g(a))(x-a)$, but in the Eq. (9) the multiplicative term $\nabla F$ is omitted, but it is not always equal to 1, right?

Page 15: "we use p(x) ≤ p(x + δ)" which is incorrect, and should be $p(x)\geq p(x+\delta)$

Page 15: "Notice that the flatness-aware item punishes the norm of gradients of samples around x + δ. Therefore, this induces ||∇logF (x + δ)||2 and ||$\nabla^{2}$logF (x + δ)|| to be 0." The implication is incorrect, the small first derivative doesn't say anything about the amplitude of the second derivate (example: $\sin x^2$)

Page 15: at the end of Appendix C, when doing all the approximations, nothing has been said about $\nabla \log p(x+\delta)$ which also needs to be close to zero, right?

**Questions:**

Q1: I would be interested in $l_0$/patch-based optimization and transferability, because it is the most applicable techniques for the real-world adversarial attacks, and the authors' thought on extensibility of their framework to this case (taking into account the differentiability problem).

Q2: Why we need the Appendix B (minmax problem complexity)? It seems like a very obvious thing and doesn't provide any insight at all

Q3: I am very interested whether the theoretic part (see my notes above) can be corrected and improved as now it seems to have a lot of mistakes.

---

> ### Author Response · Authors · 2023-11-14
> **Response (1/2)**
>
> Thank you for the efforts on our paper and your valuable suggestions.
> We have included the following response in the revised paper.
> We will upload the revised version before the end of the discussion period.
> We answer your questions point-by-point as follows.
>
> ---
>
> **Question 1:**
> I am very interested whether the theoretic part (see my notes above) can be corrected and improved as now it seems to have a lot of mistakes.
>
> **Response:**
> For the first one, I'm not sure that the Eq. (9) is correct...
> In Equation 9, we treat $L(F(\cdot),\cdot)$ as a holistic function and differentiate it, so there is no mathematical error in this expression.
> Regarding the effectiveness of linear expansion, we want to provide some discussion to support Equation 9.
> Firstly, for the domain of adversarial examples, the assumption of neural networks as approximately linear is common, as seen in [3].
> The supporting argument is that modern neural networks still exhibit a high degree of linearity, as the activation functions of neural networks often demonstrate favorable linear properties, such as ReLU.
> Furthermore, albeit empirically, assuming linearity in neural networks when studying adversarial examples often yields good results.
> Our attack results also support this.
> Lastly, within a small neighborhood, this assumption is often feasible, as per the Taylor series expansion.
> Our perturbations are typically constrained to a small magnitude to maintain the human-imperceptibility of crafted adversarial examples.
>
>
> For the second one, 'we use $p(x) \leq p(x + \delta)$ which is incorrect, and should be $p(x) \geq p(x+\delta)$'.
> We have revised this typo.
> Thanks for your careful check again.
>
> For the third one, "Notice that the flatness-aware item punishes the norm of gradients of samples around $x + \delta$. Therefore, this induces..."
> We want to clarify some things.
> In fact, according to Equation 4, our gradient penalty term is $||\nabla L(F(x+\delta+\Delta),y)||, \Delta \sim U(-b,b)$ instead of $||\nabla L(F(x+\delta),y)||$.
> The second derivative can be considered as a metric for the change rate of the first-order derivative.
> Intuitively speaking, when the penalty strength is sufficiently large, there is $||\nabla L(F(x+\delta+\Delta),y)|| \rightarrow 0$.
> Consequently, the second derivative at $x + \delta$ naturally becomes zero.
> Formally, the norm of the second derivative can be expressed by $ ||\nabla^2 L(F(x+\delta),y)|| = ||\lim_{\Delta \rightarrow 0} \{ (\nabla L(F(x+\delta+\Delta),y) - \nabla L(F(x+\delta),y)) \Delta^{-1} \}||$.
> Punishing $||\nabla L(F(x+\delta+\Delta),y)||$ indeed minimizes $||\nabla^2 L(F(x+\delta),y)||$.
>
>
> For the fourth one, at the end of Appendix C, when doing all the approximations, nothing...
> When the proxy and target models exhibit similarity, the gradients of the proxy and target model also present high similarity.
> In fact, as demonstrated in [1] (the conclusion of Section 5.1), The distance between $log p(x)$ and $log F(x)$ can control the magnitude of $||\nabla log p(x) - \nabla log F(x)||$.
> In other words, a smaller distance between $log p(x)$ and $log F(x)$ results in smaller gradient distances.
> Empirically, a well-trained model $F(x)$ typically yields a smaller $||\nabla log p(x) - \nabla log F(x)||$.
> Consequently, we are also able to impose a certain degree of penalty on $\nabla log p(x+\delta)$.
>
> ---
>
> **Question 2:**
> Page 4, the ref. to Mean Value Theorem: it'd be better to refer to some classic mathematical results (I guess they are discovered hundreds of years ago, not just 15).
>
> **Response:**
> To the best of our understanding, the origins of the Mean Value Theorem can be traced back to 1823, when its formal proof was established by mathematician Cauchy. We have updated our references accordingly.
>
> ---
>
> **Question 3:**
> Page 16, Table 5: It's not clear what "Approximation Error" exactly means: is it the overall Hessian-based additive term, or Term1, or Term2 (Equation 8)?
>
> **Response:**
> The approximation error is the overall Hessian-based additive term.
> We have revised the manuscript to enhance its clarity.
>
> ---
>
>
> **Question 4:**
> Page 19, Appendix E.8: "As shown in Figure 9, FAA produces flatter adversarial examples than FAA" -> "As shown in Figure 9, FAA produces flatter adversarial examples than RAP"?
>
> **Response:**
> Thanks for your careful check.
> This is a typo error and it is "As shown in Figure 9, FAA produces flatter adversarial examples than RAP".
> We have revised this typo.

---

> ### Author Response · Authors · 2023-11-14
> **Response (2/2)**
>
> **Question 5:**
> I would be interested in $l_0$/patch-based optimization and transferability, because it is the most applicable techniques for the real-world adversarial attacks, and the authors' thought on extensibility of their framework to this case (taking into account the differentiability problem).
>
> **Response:**
> We believe the extension of FAA to patch-based attacks is easy.
> We can directly add our gradient penalty term to the optimization targets in [4].
>
> In contrast, the constraint of L0 norm is quite hard to handle, primarily stemming from the difficulty in identifying the optimal set of pixels to optimize.
> In response, we advocate for an approximation solution as a remedy.
> Specifically, we relax L0 norm constraint by introducing an L1 regularization term (or other sparsity-inducing terms) in the optimization target (Eq. 4) to supplant the L0 norm constraint.
> This ensures the differentiability of the optimization target (Eq. 4).
> Finally，we retain perturbation elements that exert the most substantial influence on the optimization target (Eq. 4)，thereby satisfying L0 constraint.
>
> ---
>
> **Question 6:**
> Why we need the Appendix B (minmax problem complexity)? It seems like a very obvious thing and doesn't provide any insight at all.
>
> **Response:**
> To our knowledge, some researchers seem to not be very familiar with the difficulty of employing gradient descent algorithms to address the bi-level optimization problem.
> Thus, we provide an example to supplement this.
>
> ---
>
> Reference:
>
> [1] On the Second Mean-Value Theorem of the Integral Calculus, 1909
>
> [2] On the Robustness of Split Learning against Adversarial Attacks, ECAI 2023.
>
> [3] Explaining and harnessing adversarial examples, ICLR 2014.
>
> [4] Adversarial T-shirt! Evading Person Detectors in A Physical World, ECCV 2020.

---

> > ### Author Response · Authors · 2023-11-15
> > **Discussion Inquiry**
> >
> > Dear Reviewer,
> >
> > We thank you for the precious review time and valuable comments. We have provided responses to your question and the weakness you mentioned. We hope this can address your concerns.
> >
> > We hope to further discuss with you whether or not your concerns have been addressed appropriately. Please let us know if you have additional questions or comments. We look forward to hearing from you soon.
> >
> > Best regards,
> >
> > Authors

---

> > > ### Author Response · Authors · 2023-11-17
> > > **Looking forward to your feedback**
> > >
> > > Dear Reviewer,
> > >
> > > Sorry to bother you again. With the discussion phase nearing its end, we would like to know whether the responses have addressed your concerns.
> > >
> > > Should this be the case, we are encouraged that you raise the final rating to reflect this.
> > >
> > > If there are any remaining concerns that have led to a negative evaluation, please let us know. We are more than willing to engage in further discussion and address any remaining concerns to the best of our abilities.
> > >
> > > We are looking forward to your reply. Thank you for your efforts on this paper.
> > >
> > > Best regards,
> > >
> > > Authors

---

> > > > ### Author Response · Authors · 2023-11-17
> > > > **Discussion Inquiry**
> > > >
> > > > Dear Reviewer,
> > > >
> > > > We have submitted our response to your questions and are anticipating whether it addresses the questions you raised.
> > > >
> > > > Could you spare some time to read our response? We appreciate your effort on this paper.
> > > >
> > > > Best regards,
> > > >
> > > > Authors

---

### Official Review · Reviewer_DvXX · 2023-11-01

**Soundness:** 3 good
**Presentation:** 3 good
**Contribution:** 3 good
**Rating:** 8
**Confidence:** 4

**Summary:**

The paper presents an adversarial attack algorithm that leverages flat loss regions to generate transferable adversarial examples. Current methods use input regularization to generate better transferable adversarial examples. In this work, the authors instead derive a flatness-aware regularization term. Further, they propose a Hessian approximation for the gradients. The approach is shown to outperform existing transfer attacks.

**Strengths:**

1. The approach is well-motivated and is intuitive. The overall presentation is good, and the paper is well written.
2. While similar approaches have been applied to defenses, flatness aware attacks are an interesting application.
3. The paper also shows effective results for real world image classifiers as well as a variety of robust and non-robust models.
4. The attack is significantly successful even when the proxy is not adversarially trained.

**Weaknesses:**

It might be useful to evaluate the approach for flatness-aware adversarial defenses like TRADES (Zhang et al, ICML 2019), SAM-AT [1], and ATEnt [2]. These methods leverage sharpness aware losses to introduce flat loss landscapes with respect to the input, and could possibly behave differently.

[1] Wei, Zeming, Jingyu Zhu, and Yihao Zhang. "On the Relation between Sharpness-Aware Minimization and Adversarial Robustness.", ADvML Workshop at ICML 2023.
[2] Jagatap, Gauri, et al. "Adversarially robust learning via entropic regularization." Frontiers in artificial intelligence 4 (2022): 780843.

**Questions:**

See weaknesses above.